DOI: 10.1038/s41467-017-02380-9　　**OPEN**

Corrected: Publisher correction

# Re-analysis of public genetic data reveals a rare X-chromosomal variant associated with type 2 diabetes

Sílvia Bonàs-Guarch et al.#

The reanalysis of existing GWAS data represents a powerful and cost-effective opportunity to gain insights into the genetics of complex diseases. By reanalyzing publicly available type 2 diabetes (T2D) genome-wide association studies (GWAS) data for 70,127 subjects, we identify seven novel associated regions, five driven by common variants (*LYPLAL1*, *NEUROG3*, *CAMKK2*, *ABO*, and *GIP* genes), one by a low-frequency (*EHMT2*), and one driven by a rare variant in chromosome Xq23, rs146662075, associated with a twofold increased risk for T2D in males. rs146662075 is located within an active enhancer associated with the expression of Angiotensin II Receptor type 2 gene (*AGTR2*), a modulator of insulin sensitivity, and exhibits allelic specific activity in muscle cells. Beyond providing insights into the genetics and pathophysiology of T2D, these results also underscore the value of reanalyzing publicly available data using novel genetic resources and analytical approaches.

During the last decade, hundreds of genome-wide association studies (GWAS) have been performed with the aim of providing a better understanding of the biology of complex diseases, improving their risk prediction, and ultimately discovering novel therapeutic targets[1]. However, the majority of the published GWAS have only reported primary findings, which generally explain a small fraction of the estimated heritability. To examine the missing heritability, most strategies involve generating new genetic and clinical data. Very rarely are new studies based on the revision and reanalysis of existing genetic data by applying more powerful analytic techniques and resources after the primary GWAS findings are published. These cost-effective reanalysis strategies are now possible, given emerging (1) data-sharing initiatives with large amounts of primary genetic data for multiple human genetic diseases, as well as (2) new and improved GWAS methodologies and resources. Notably, genotype imputation with novel sequence-based reference panels can now substantially increase the genetic resolution of GWASs from previously genotyped data sets[2], reaching good-quality imputation of low frequency (minor allele frequency [MAF]: $0.01 \leq MAF < 0.05$) and rare variants (MAF < 0.01), increasing the power to identify novel associations, and fine map the known ones. Moreover, the availability of publicly available primary genetic data allows the homogeneous integration of multiple data sets from different origins providing more accurate meta-analysis results, particularly at the low ranges of allele frequency. Finally, the vast majority of reported GWAS analyses omits the X chromosome, despite representing 5% of the genome and coding for more than 1,500 genes[3]. The reanalysis of publicly available data also enables interrogation of this chromosome.

We hypothesized that a unified reanalysis of multiple publicly available data sets, applying homogeneous standardized quality control (QC), genotype imputation, and association methods, as well as novel and denser sequence-based reference panels for imputation would provide new insights into the genetics and the pathophysiology of complex diseases. To test this hypothesis, we focused this study on type 2 diabetes (T2D), one of the most prevalent complex diseases for which many GWAS have been performed during the past decade[4]. These studies have allowed the identification of more than 100 independent loci, most of them driven by common variants, with a few exceptions[5]. Despite these efforts, there is still a large fraction of genetic heritability hidden in the data, and the role of low-frequency variants, although recently proposed to be minor[6], has still not been fully explored. The availability of large T2D genetic data sets in combination with larger and more comprehensive genetic variation reference panels[2], provides the opportunity to impute a significantly increased fraction of low-frequency and rare variants, and to study their contribution to the risk of developing this disease. This strategy also allows us to fine map known associated loci, increasing the chances of finding causal variants and understanding their functional impact. We therefore gathered publicly available T2D GWAS cohorts with European ancestry, comprising a total of 13,857 T2D cases and 62,126 controls, to which we first applied harmonization and quality control protocols covering the whole genome (including the X chromosome). We then performed imputation using 1000 Genomes Project (1000G)[7] and UK10K[2] reference panels, followed by association testing. By using this strategy, we identified novel associated regions driven by common, low-frequency and rare variants, fine mapped and functionally annotated the existing and novel ones, allowing us to describe a regulatory mechanism disrupted by a novel rare and large-effect variant identified at the X chromosome.

## Results

**Overall analysis strategy.** As shown in Fig. 1, we first gathered all T2D case-control GWAS individual-level data that were available through the EGA and dbGaP databases (i.e., Gene Environment-Association Studies [GENEVA], Wellcome Trust Case Control Consortium [WTCCC], Finland–United States Investigation of NIDDM Genetics [FUSION], Resource for Genetic Epidemiology Research on Aging [GERA], and Northwestern NuGENE project [NuGENE]). We harmonized these cohorts, applied standardized quality control procedures, and filtered out low-quality variants and samples (Methods and Supplementary Notes). After this process, a total of 70,127 subjects (70KforT2D, 12,931 cases, and 57,196 controls, Supplementary Data 1) were retained for downstream analysis. Each of these cohorts was then imputed to the 1000G and UK10K reference panels using an integrative method, which selected the results from the reference panel that provided the highest accuracy for each variant, according to IMPUTE2 info score (Methods). Finally, the results from each of these cohorts were meta-analyzed (Fig. 1), obtaining a total of 15,115,281 variants with good imputation quality (IMPUTE2 info score $\geq 0.7$, MAF $\geq 0.001$, and $I^2$ heterogeneity score < 0.75), across 12,931 T2D cases and 57,196 controls. Of these, 6,845,408 variants were common (MAF $\geq 0.05$), 3,100,848 were low-frequency ($0.01 \leq MAF < 0.05$), and 5,169,025 were rare ($0.001 \leq MAF < 0.01$). Merging the imputation results derived from the two reference panels substantially improved the number of good-quality imputed variants, particularly within the low-frequency and rare spectrum, compared to the imputation results obtained with each of the panels separately. For example, a set of 5,169,025 rare variants with good quality was obtained after integrating 1000G and UK10K results, while only 2,878,263 rare variants were imputed with 1000G and 4,066,210 with UK10K (Supplementary Fig. 1A). This strategy also allowed us to impute 1,357,753 indels with good quality (Supplementary Fig. 1B).

To take full advantage of publicly available genetic data, we used three main meta-analytic approaches to adapt to the three most common strategies for genetic data sharing: individual-level genotypes, summary statistics, and single-case queries through the Type 2 Diabetes Knowledge Portal (T2D Portal) (http://www.type2diabetesgenetics.org/). We first meta-analyzed all summary statistics results from the DIAGRAM trans-ancestry meta-analysis[8] (26,488 cases and 83,964 controls), selecting 1,918,233 common variants (MAF $\geq 0.05$), mostly imputed from HapMap, with the corresponding fraction of non-overlapping samples in our 70KforT2D set, i.e. the GERA and the NuGENE cohorts, comprising a total of 7,522 cases and 50,446 controls (Fig. 1, Supplementary Data 1). Second, the remaining variants (13,197,048), which included mainly non-HapMap variants (MAF < 0.05) or variants not tested above, were meta-analyzed using all five cohorts from the 70KforT2D resource (Supplementary Data 1). Finally, low-frequency variants located in coding regions and with $p \leq 1 \times 10^{-4}$ were meta-analyzed using the non-overlapping fraction of samples with the data from the T2D Portal through the interrogation of exome array and whole-exome sequence data from ~80,000 and ~17,000 individuals, respectively[6].

**Pathway and functional enrichment analysis.** To explore whether our results recapitulate the pathophysiology of T2D, we performed gene-set enrichment analysis with all the variants with $p \leq 1 \times 10^{-5}$ using DEPICT[9] (Methods). This analysis showed enrichment of genes expressed in pancreas (ranked first in tissue enrichment analysis, $p = 7.8 \times 10^{-4}$, FDR < 0.05, Supplementary Data 2) and cellular response to insulin stimulus (ranked second in gene-set enrichment analysis, $p = 3.9 \times 10^{-8}$, FDR = 0.05,

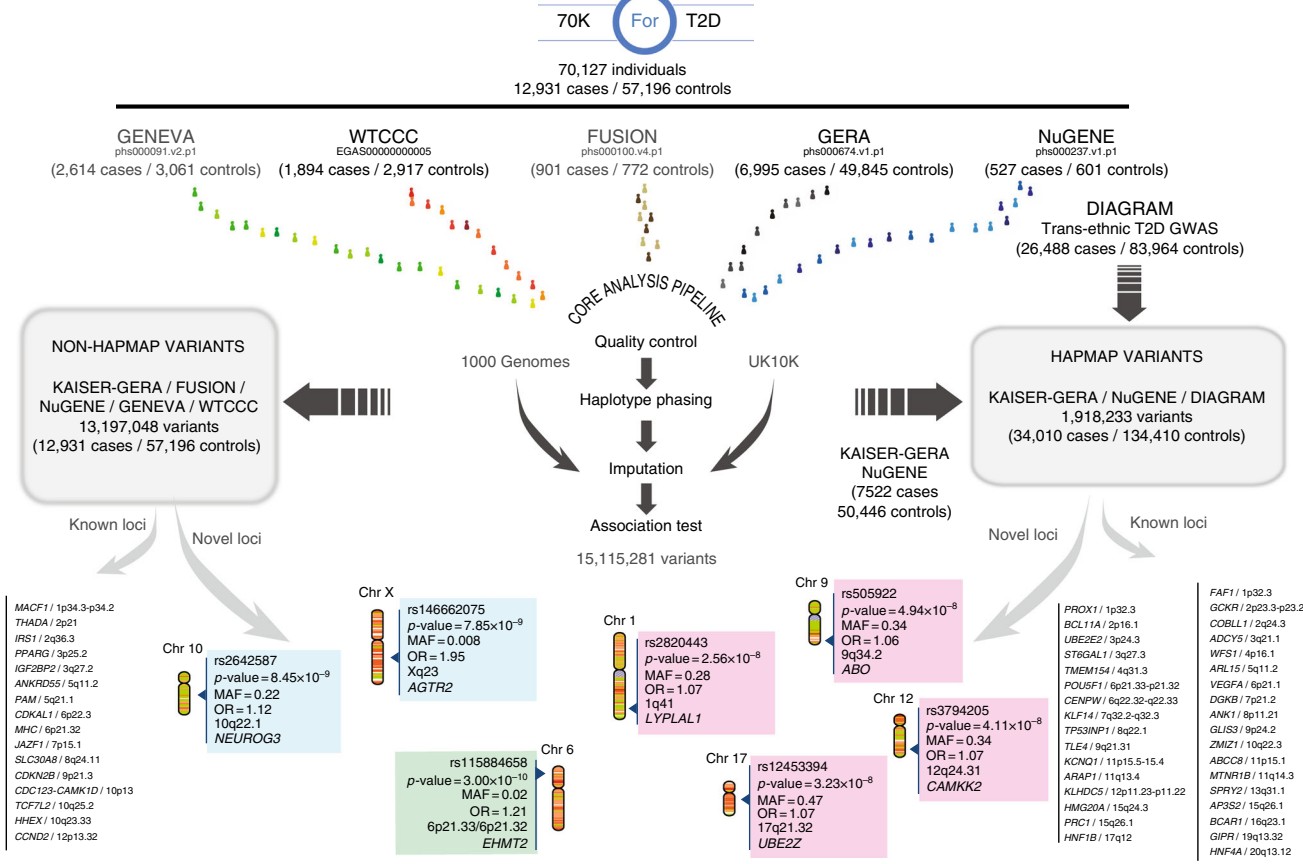

**Fig. 1** Discovery and replication strategy. Publicly available GWAS datasets representing a total of 12,931 cases and 57,196 controls (70KforT2D) were first quality controlled, phased, and imputed, using 1000G and UK10K separately. For those variants that were present in the DIAGRAM trans-ethnic meta-analysis, we used the summary statistics to meta-analyze our results with the cohorts that had no overlap with any of the cohorts included in the DIAGRAM trans-ethnic meta-analysis. With this first meta-analysis, we discovered four novel loci (within magenta panels). For the rest of the variants, we meta-analyzed all the 70KforT2D data sets, which resulted in two novel loci (in blue panels). All the variants that were coding and showed a *p*-value of $\leq 1 \times 10^{-4}$ were tested for replication by interrogating the summary statistics in the Type 2 Diabetes Knowledge Portal (T2D Portal) (http://www.type2diabetesgenetics.org/). This uncovered a novel low-frequency variant in the *EHMT2* gene (highlighted with a green panel)

Supplementary Data 3, Supplementary Fig. 2, Supplementary Fig. 3), in concordance with the current knowledge of the molecular basis of T2D.

In addition, variant set enrichment analysis of the T2D-associated credible sets across regulatory elements defined in isolated human pancreatic islets showed a significant enrichment for active regulatory enhancers (Supplementary Fig. 4), suggesting that causal SNPs within associated regions have a regulatory function, as previously reported[10].

**Fine-mapping and functional characterization of T2D loci.** The three association strategies allowed us to identify 57 genome-wide significant associated loci ($p \le 5 \times 10^{-8}$), of which seven were not previously reported as associated with T2D (Table 1). The remaining 50 loci have been previously reported and included, for example, two low-frequency variants recently discovered in Europeans, one located within one of the *CCND2* introns (rs76895963), and a missense variant within the *PAM*[5] gene. Furthermore, we confirmed that the magnitude and direction of the effect of all the associated variants ($p \le 0.001$) were highly consistent with those reported previously ($\rho = 0.92$, $p = 1 \times 10^{-248}$, Supplementary Fig. 5). In addition, the direction of effect was consistent with all 139 previously reported variants, except three that were discovered in east and south Asian populations (Supplementary Data 4).

The high coverage of genetic variation ascertained in this study allowed us to fine-map known and novel loci, providing more candidate causal variants for downstream functional interpretations. We constructed 99% credible variant sets[11] for each of these loci, i.e. the subset of variants that have, in aggregate, 99% probability of containing the true causal variant for all 57 loci (Supplementary Data 5). As an important improvement over previous T2D genetic studies, we identified small structural variants within the credible sets, consisting mostly of insertions and deletions between 1 and 1,975 nucleotides. In fact, out of the 8,348 variants included within the credible sets for these loci, 927 (11.1%) were indels, of which 105 were genome-wide significant (Supplementary Data 6). Interestingly, by integrating imputed results from 1000G and UK10K reference panels, we gained up to 41% of indels, which were only identified by either one of the two reference panels, confirming the advantage of integrating the results from both reference panels. Interestingly, 15 of the 71 previously reported loci that we replicated ($p \le 5.3 \times 10^{-4}$ after correcting for multiple testing) have an indel as the top variant, highlighting the potential role of this type of variation in the susceptibility for T2D. For example, within the *IGF2BP2* intron, a well-established and functionally validated locus for T2D[12], we found that 12 of the 57 variants within its 99% credible set correspond to indels with genome-wide significance ($5.6 \times 10^{-16} < p < 2.4 \times 10^{-15}$), which collectively represented 18.4% posterior probability of being causal.

**Table 1 Novel T2D-associated loci**

| Novel Locus | Chr | rsID–-Risk Allele | OR (95% CI) P-value | | | MAF |
|---|---|---|---|---|---|---|
| | | | Stage1 Discovery Meta-analysis | Stage2 Replication Meta-analysis | Stage1 + Stage2 Combined Meta-analysis | |
| LYPLAL1/ZC3H11B (1q41) | 1 | rs2820443-T | 1.08 (1.04–1.13) $2.94 \times 10^{-4}$ [a] | 1.06 (1.03–1.09) $2.10 \times 10^{-5}$ [b] | 1.07 (1.04–1.09) $2.56 \times 10^{-8}$ [c] | 0.28 |
| EHMT2 (6p21.33–p21.32) | 6 | rs115884658-A | 1.34 (1.18–1.53) $1.00 \times 10^{-5}$ [a] | 1.17 (1.09–1.26) $2.90 \times 10^{-6}$ [c, d] | 1.21 (1.14–1.29) $3.00 \times 10^{-10}$ [c] | 0.02 |
| ABO (9q34.2) | 9 | rs505922-C | 1.07 (1.03–1.11) $6.93 \times 10^{-4}$ [a] | 1.06 (1.03–1.09) $1.90 \times 10^{-5}$ [b] | 1.06 (1.04–1.09) $4.94 \times 10^{-8}$ [c] | 0.34 |
| NEUROG3 (10q22.1) | 10 | rs2642587-G | 1.12 (1.08–1.16) $8.45 \times 10^{-9}$ [e] | - | - | 0.22 |
| CAMKK2 (12q24.31) | 12 | rs3794205-G | 1.09 (1.05–1.14) $4.18 \times 10^{-5}$ [a] | 1.06 (1.03–1.09) $1.60 \times 10^{-4}$ [b] | 1.07 (1.04–1.10) $4.11 \times 10^{-8}$ [c] | 0.32 |
| CALCOCO2/ATP5G1/ UBE2Z/SNF8/GIP (17q21.32) | 17 | rs12453394-A | 1.08 (1.04–1.12) $7.86 \times 10^{-5}$ [a] | 1.07 (1.03–1.11) $9.60 \times 10^{-5}$ [b] | 1.07 (1.05–1.10) $3.23 \times 10^{-8}$ [c] | 0.47 |
| AGTR2 (Xq23) | X | rs146662075-T | 3.09 (2.06–4.60) $3.24 \times 10^{-8}$ [f] | 1.57 (1.19–2.07) $1.42 \times 10^{-3}$ [g] | 1.95 (1.56–2.45) $7.85 \times 10^{-9}$ | 0.008 |

Chr chromosome, OR odds ratio, MAF minor allele frequency
[a]Imputed based public GWAS discovery meta-analysis (NuGENE + GERA cohort, 7,522 cases and 50,446 controls)
[b]Transancestry DIAGRAM Consortium (26,488 cases and 83,964 controls)[c]Meta P-value estimated using a weighted Z-score method due to unavailable SE information from Stage 2 replication cohorts[d]T2D Diabetes Genetic Portal (Exome-Chip + Exome Sequencing, 35,789 cases and 56,738 controls)[e]Full imputed based public GWAS meta-analysis (NuGENE + GERA cohort + GENEVA + FUSION + WTCCC, 12,931 cases and 57,196 controls)
[f]70KforT2D Men Cohort (GERA cohort + GENEVA + FUSION, 5,277 cases and 15,702 controls older than 55 years)
[g]Replication Men Cohort SIGMA UK10K imputation + InterAct + Danish Cohort (case control and follow-up) + Partners Biobank + UK Biobank (18,370 cases and 88,283 controls older than 55 years and OGTT > 7.8 mmol l$^{-1}$, when available)

To prioritize causal variants within all the identified associated loci, we annotated their corresponding credible sets using the Variant Effector Predictor (VEP) for coding variants[13] (Supplementary Data 7), and the Combined Annotation-Dependent Depletion (CADD)[14] and LINSIGHT[15] tools for non-coding variation (Supplementary Data 8 and 9). In addition, we tested the effect of all variants on expression across multiple tissues by interrogating GTEx[16] and RNA-sequencing gene expression data from pancreatic islets[17].

**Novel T2D-associated loci driven by common variants.** Beyond the detailed characterization of the known T2D-associated regions, we also identified seven novel loci, among which, five were driven by common variants with modest effect sizes (1.06 < OR < 1.12; Table 1, Fig. 2, Supplementary Fig. 6 and 7).

Within the first novel T2D-associated locus in chromosome 1q41 (LYPLAL1-ZC3H11B, rs2820443, OR = 1.07 [1.04–1.09], p = 2.6 × 10$^{-8}$), several variants have been previously associated with waist-to-hip ratio, height, visceral adipose fat in females, adiponectin levels, fasting insulin, and non-alcoholic fatty liver disease[18–23]. Among the genes in this locus, LYPLAL1, which encodes for lysophospholipase-like 1, appears to be the most likely effector gene, as it has been found to be downregulated in mouse models of diet-induced obesity and upregulated during adipogenesis[24].

Second, a novel locus at chromosome 9q34.2 region (ABO, rs505922, OR = 1.06 [1.04–1.09], p = 4.9 × 10$^{-8}$) includes several variants that have been previously associated with other metabolic traits. For example, the variant rs651007, in linkage disequilibrium (LD) with rs505922 (r$^2$ = 0.507), has been shown to be associated with fasting glucose[25], and rs514659 (r$^2$ with top = 1) is associated with an increased risk for cardiometabolic disorders[26]. One of the variants within the credible set was the single base-pair frame-shift deletion defining the blood group O[27]. In concordance with previous results that linked O blood type with a lower risk of developing T2D[28], the frame-shift deletion determining the blood group type O was associated with

a protective effect for T2D in our study (rs8176719, p = 3.4 × 10$^{-4}$, OR = 0.95 [0.91–0.98]). In addition, several variants within this credible set are associated with the expression of the ABO gene in multiple tissues including skeletal muscle, adipose tissue, and pancreatic islets (Supplementary Data 9, Supplementary Data 10).

Third, a novel locus at chromosome 10q22.1 locus (NEUROG3/ COL13A1/RPL5P26, rs2642587, OR = 1.12 [1.08–1.16], p = 8.4 × 10$^{-9}$) includes NEUROG3 (Neurogenin3), which is an essential regulator of pancreatic endocrine cell differentiation[29]. Mutations in this gene have been reported to cause permanent neonatal diabetes, but a role of this gene in T2D has not been yet reported[30].

The lead common variant of the fourth novel locus at chromosome 12q24.31 (rs3794205, OR = 1.07 [1.04–1.10], p = 4.1 × 10$^{-8}$) lies within an intron of the CAMKK2 gene, previously implicated in cytokine-induced beta-cell death[31]. However, other variants within the corresponding credible set could also be causal, such as a missense variant within the P2RX7, a gene previously associated with glucose homeostasis in humans and mice[32], or another variant (rs11065504, r$^2$ with lead variant = 0.81) found to be associated with the regulation of the P2RX4 gene in tibial artery and in whole blood, according to GTEx (Supplementary Data 9).

The fifth novel locus driven by common variants is located within 17q21.32 (rs12453394, OR = 1.07 [1.05–1.10], p = 3.23 × 10$^{-8}$). It includes three missense variants located within the CALCOCO2, SNF8, and GIP genes. GIP encodes for glucose-dependent insulinotropic peptide, a hormonal mediator of enteral regulation of insulin secretion[33]. Variants in the GIP receptor (GIPR) have been previously associated with insulin response to oral glucose challenge and beta-cell function[34], proposing GIP as a plausible candidate effector gene of this locus[35].

**A new T2D signal driven by a low-frequency variant.** Furthermore, we selected all low-frequency (0.01 ≤ MAF < 0.05) variants with p ≤ 1 × 10$^{-4}$ in the 70KforT2D meta-analysis that

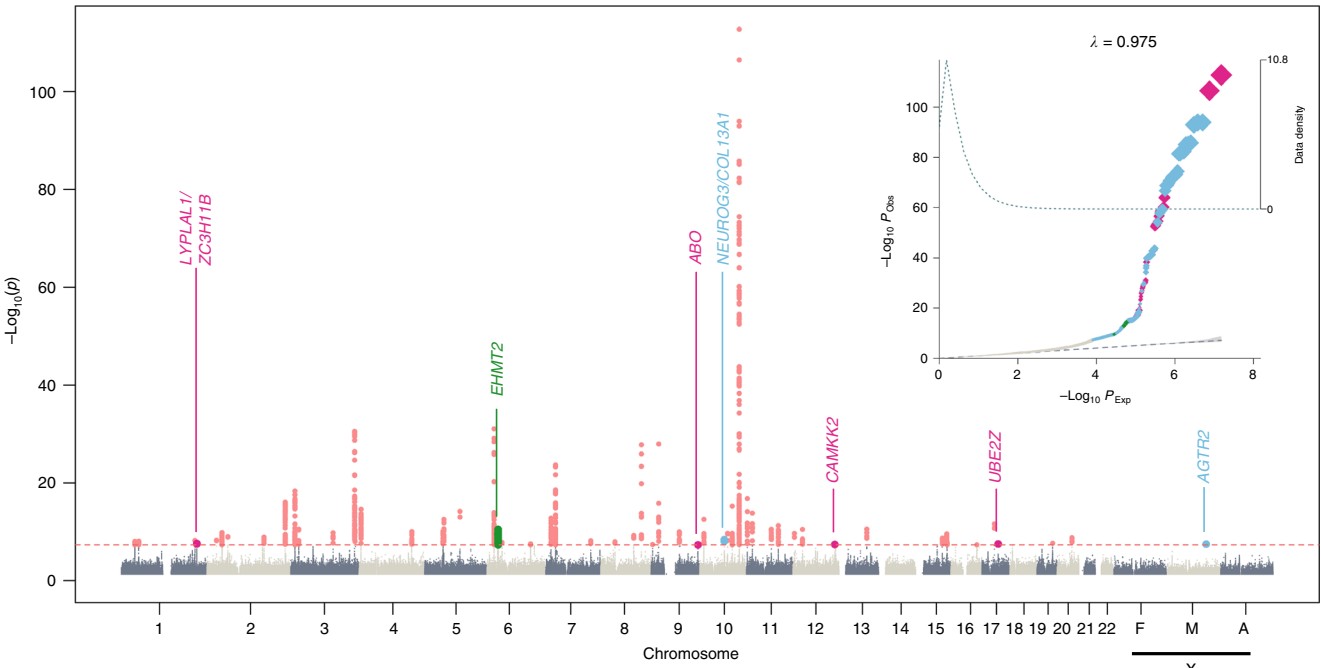

**Fig. 2** Manhattan and quantile–quantile plot (QQ-plot) of the discovery and replication genome-wide meta-analysis. The upper corner represents the QQ-plot. Expected $-\log_{10}$ p-values under the null hypothesis are represented in the x axis, while observed $-\log_{10}$ p-values are represented in the y axis. Observed p-values were obtained according to the suitable replication dataset used (as shown in Fig. 1) and were depicted using different colors. HapMap variants were meta-analyzed using the trans-ethnic summary statistics from the DIAGRAM study and our meta-analysis based on the Genetic Epidemiology Research on Aging (GERA) cohort and the northwestern NuGENE project, and that resulted in novel associations depicted in magenta. The rest of non-HapMap variants meta-analyzed using the full 70KforT2D cohort are represented in gray, and the fraction of novel GWAS-significant variants is highlighted in light blue. Coding low-frequency variants meta-analyzed using the 70KforT2D and the T2D Portal data that resulted in novel GWAS-significant associations are depicted in green. The shaded area of the QQ-plot indicates the 95% confidence interval under the null and a density function of the distribution of the p-values was plotted using a dashed line. The $\lambda$ is a measure of the genomic inflation and corresponds to the observed median $\chi 2$ test statistic divided by the median expected $\chi^2$ test statistic under the null hypothesis. The Manhattan plot, representing the $-\log_{10}$ p-values, was colored as explained in the QQ-plot. All known GWAS-significant associated variants within known T2D genes are also depicted in red. X chromosome results for females (F), males (M), and all individuals (A) are also included

were annotated as altering protein sequences, according to VEP. This resulted in 15 coding variants that were meta-analyzed with exome array and whole-exome sequencing data from a total of ~97,000 individuals[6] after excluding the overlapping cohorts between the different data sets. This analysis highlighted a novel genome-wide association driven by a low-frequency missense variant (Ser58Phe) within the *EHMT2* gene at chromosome 6p21.33 (rs115884658, OR = 1.21 [1.14–1.29], $p = 3.00 \times 10^{-10}$; Fig. 2, Supplementary Figures 6 and 7). *EHMT2* is involved in the mediation of FOXO1 translocation induced by insulin[36]. Since this variant is less than 1 Mb away from *HLA-DQA1*, a locus reported to be associated with T2D[37], we performed a series of reciprocal conditional analyses and excluded the possibility that our analysis was capturing previously reported T2D[8, 37] or T1D[38–40] signals (Supplementary Data 11). Beyond this missense *EHMT2* variant, other low-frequency variants within the corresponding credible set may also be causal. For example, rs115333512 ($r^2$ with lead variant = 0.28) is associated with the expression of *CLIC1* in several tissues according to GTEx (multitissue meta-analysis $p = 8.9 \times 10^{-16}$, Supplementary Data 9). In addition, this same variant is associated with the expression of the first and second exon of the *CLIC1* mRNA in pancreatic islet donors ($p(\text{exon } 1) = 1.4 \times 10^{-19}$, $p(\text{exon } 2) = 1.9 \times 10^{-13}$, Supplementary Data 10). Interestingly, *CLIC1* has been reported as a direct target of metformin by mediating the antiproliferative effect of this drug in human glioblastoma[41]. All these findings support *CLIC1,* as an additional possible effector transcript, likely driven by rs115333512.

**A novel rare X chromosome variant associated with T2D**. Similar to other complex diseases, the majority of published large-scale T2D GWAS studies have omitted the analysis of the X chromosome, with the notable exception of the identification of a T2D-associated region near the *DUSP9* gene in 2010[42]. To fill this gap, we tested the X chromosome genetic variation for association with T2D. To account for heterogeneity of the effects and for the differences in imputation performance between males and females, the association was stratified by sex and tested separately, and then meta-analyzed. This analysis was able to replicate the *DUSP9* locus, not only through the known rs5945326 variant (OR = 1.15, $p = 0.049$), but also through a three-nucleotide deletion located within a region with several promoter marks in liver (rs61503151 [GCCA/G], OR = 1.25, $p = 3.5 \times 10^{-4}$), and in high LD with the first reported variant ($r^2 = 0.62$). Conditional analyses showed that the originally reported variant was no longer significant (OR = 1.01, $p = 0.94$) when conditioning on the newly identified variant, rs61503151. On the other hand, when conditioning on the previously reported variant, rs5945326, the effect of the newly identified indel remained significant and with a larger effect size (OR = 1.33, $p = 0.003$), placing this deletion, as a more likely candidate causal variant for this locus (Supplementary Data 14).

In addition, we identified a novel genome-wide significant signal in males at the Xq23 locus driven by a rare variant (rs146662075, MAF = 0.008, OR = 2.94 [2.00–4.31], $p = 3.5 \times 10^{-8}$; Fig. 3a). Two other variants in LD with the top variant, rs139246371 (chrX:115329804, OR = 1.65, $p = 3.5 \times 10^{-5}$, $r^2 = $

0.37 with the top variant) and rs6603744 (chrX:115823966, OR = 1.28, $p = 1.7 \times 10^{-4}$, $r^2 = 0.1$ with the top variant), comprised the 99% credible set and supported the association. We tested in detail the accuracy of the imputation for the rs146662075 variant by comparing the imputed results from the same individuals genotyped by two different platforms (Methods) and found that the imputation was highly accurate in males only when using UK10K, but not in females, nor when using 1000G ($R^2_{[UK10K,males]} = 0.94$, $R^2_{[UK10K,females]} = 0.66$, $R^2_{[1000G,males]} = 0.62$, and $R^2_{[1000G,females]} = 0.43$; Supplementary Fig. 8). Whether this association is specific to men, or whether it also affects female carriers, remains to be clarified with datasets that allow accurate imputation on females, or with direct genotyping or sequencing.

To further validate and replicate this association, we next analyzed four independent data sets (SIGMA[6], INTERACT[43], Partners Biobank[44], and UK Biobank[45]), by performing imputation with the UK10K reference panel. In addition, a fifth cohort was genotyped de novo for the rs146662075 variant in several Danish sample sets. The initial meta-analysis, including the five replication data sets did not reach genome-wide significance (OR = 1.57, $p = 1.2 \times 10^{-5}$; Supplementary Fig. 9A), and revealed a strong degree of heterogeneity (heterogeneity $p_{het} = 0.004$), which appeared to be driven by the replication cohorts.

As a complementary replication analysis, within one of the case-control studies, there was a nested prospective cohort study, the Inter99, which consisted of 1,652 nondiabetic male subjects genotyped for rs146662075, of which 158 developed T2D after 11 years of follow-up. Analysis of incident diabetes in this cohort confirmed the association with the same allele, as previously seen in the case-control studies, with carriers of the rare T allele having increased risk of developing incident diabetes, compared to the C carriers (Cox-proportional hazards ratio (HR) = 3.17 [1.3–7.7], $p = 0.011$, Fig. 3b). Nearly 30% of carriers of the T risk allele developed incident T2D during 11 years of follow-up, compared to only 10% of noncarriers.

To understand the strong degree of heterogeneity observed after adding the replication datasets, we compared the clinical and demographic characteristics of the discovery and replication cohorts, and found that the majority of the replication datasets contained control subjects that were significantly younger than 55 years, the average age at the onset of T2D reported in this study and in Caucasian populations[46]. This was particularly clear for the Danish cohort (age controls [95%CI] = 46.9 [46.6–47.2]) vs. age cases [95%CI] = 60.7 [60.4–61.0]) and for INTERACT (age controls [95%CI] = 51.7 [51.4–52.1] vs. age cases [95%CI] = 54.8 [54.6–55.1]; Supplementary Fig. 10). Given the supporting results with the Inter99 prospective cohort, we performed an additional analysis using a stricter definition of controls, to minimize the presence of prediabetics or individuals that may further develop diabetes after reaching the average age at the onset. For this, we applied two additional exclusion criteria: (i) subjects younger than 55 years and (ii), when possible, excluding individuals with measured 2-h plasma glucose values during oral glucose tolerance test (OGTT) above 7.8 mmol l$^{-1}$, a threshold employed to identify impaired glucose tolerance (prediabetes)[47], or controls with family history of T2D, both being strong risk factors for developing T2D. While the application of the first filter alone did not yield genome-wide significant results (Supplementary Fig. 9B), upon excluding individuals with prediabetes or a family history of T2D, the replication results were significant and consistent with the initial discovery results (OR = 1.57 [1.19–2.07], $p = 0.0014$). The combined analysis of the discovery and replication cohorts resulted in genome-wide significance, confirming the association of rs146662075 with T2D (OR = 1.95 [1.56–2.45], $p = 7.8 \times 10^{-9}$, Fig. 3c).

**Allele-specific enhancer activity of the rs146662075 variant.** We next explored the possible molecular mechanism behind this association, by using different genomic resources and experimental approaches. The credible set of this region contained three variants, with the leading SNP alone (rs146662075), showing 78% posterior probability of being causal (Supplementary Fig. 7, Supplementary Data 5), as well as the highest CADD (scaled C-score = 15.68; Supplementary Data 8), and LINSIGHT score (Supplementary Data 9). rs146662075 lies within a chromosomal region enriched in regulatory (DNase I) and active enhancer (H3K27ac) marks, between the *AGTR2* (at 103 kb) and the *SLC6A14* (at 150 kb) genes. The closest gene *AGTR2*, which encodes for the angiotensin II receptor type 2, has been previously associated with insulin secretion and resistance[48–50]. From the analysis of available epigenomic data sets[51], we found no evidences of H3K27ac or other enhancer regulatory marks in human pancreatic islets; whereas a significant association was observed between the presence of H3K27ac enhancer marks and the expression of *AGTR2* across multiple tissues (Fisher test $p = 4.45 \times 10^{-3}$), showing the highest signal of both H3K27ac and *AGTR2* RNA-seq expression, but not with other genes from the same topologically associated domain (TAD), in fetal muscle (Fig. 4a; Supplementary Figure 11).

We next studied whether the region encompassing the rs146662075 variant could act as a transcriptional enhancer and whether its activity was allele-specific. For this, we linked the DNA region with either the T (risk) or the C (non-risk) allele, to a minimal promoter and performed luciferase assays in a mouse myoblast cell line. The luciferase analysis showed an average 4.4-fold increased activity for the disease-associated T allele, compared to the expression measured with the common C allele, suggesting an activating function of the T allele, or a repressive function of the C allele (Fig. 4b). Consistent with these findings, electrophoretic mobility shift assays using nuclear protein extracts from mouse myoblast cell lines, differentiated myotubes, and human fetal muscle cell line, revealed sequence-specific binding activity of the C allele, but not the rare T allele (Fig. 4c). Overall, these data indicate that the risk T allele prevents the binding of a nuclear protein that is associated with decreased activity of an *AGTR2*-linked enhancer.

## Discussion

Through harmonizing and reanalyzing publicly available T2D GWAS data, and performing genotype imputation with two whole-genome sequence-based reference panels, we are able to perform deeper exploration of the genetic architecture of T2D. This strategy allowed us to impute and test for association with T2D more than 15 million of high-quality imputed variants, including low-frequency, rare, and small insertions and deletions, across chromosomes 1–22 and X.

The reanalysis of these data confirmed a large fraction of already-known T2D loci, and identified novel potential causal variants by fine mapping and functionally annotating each locus.

This reanalysis also allowed us to identify seven novel associations, five driven by common variants in or near *LYPLAL1*, *NEUROG3*, *CAMKK2*, *ABO*, and *GIP*; a low-frequency variant in *EHMT2*, and a rare variant in the X chromosome. This rare variant identified in Xq23 chromosome was located near the *AGTR2* gene, and showed nearly twofold increased risk for T2D in males, which represents, to our knowledge, the largest effect size identified so far in Europeans, and a magnitude similar to other variants with large effects identified in other populations[52, 53].

Our study complemented other efforts that also aim at unraveling the genetics behind T2D through the generation of new

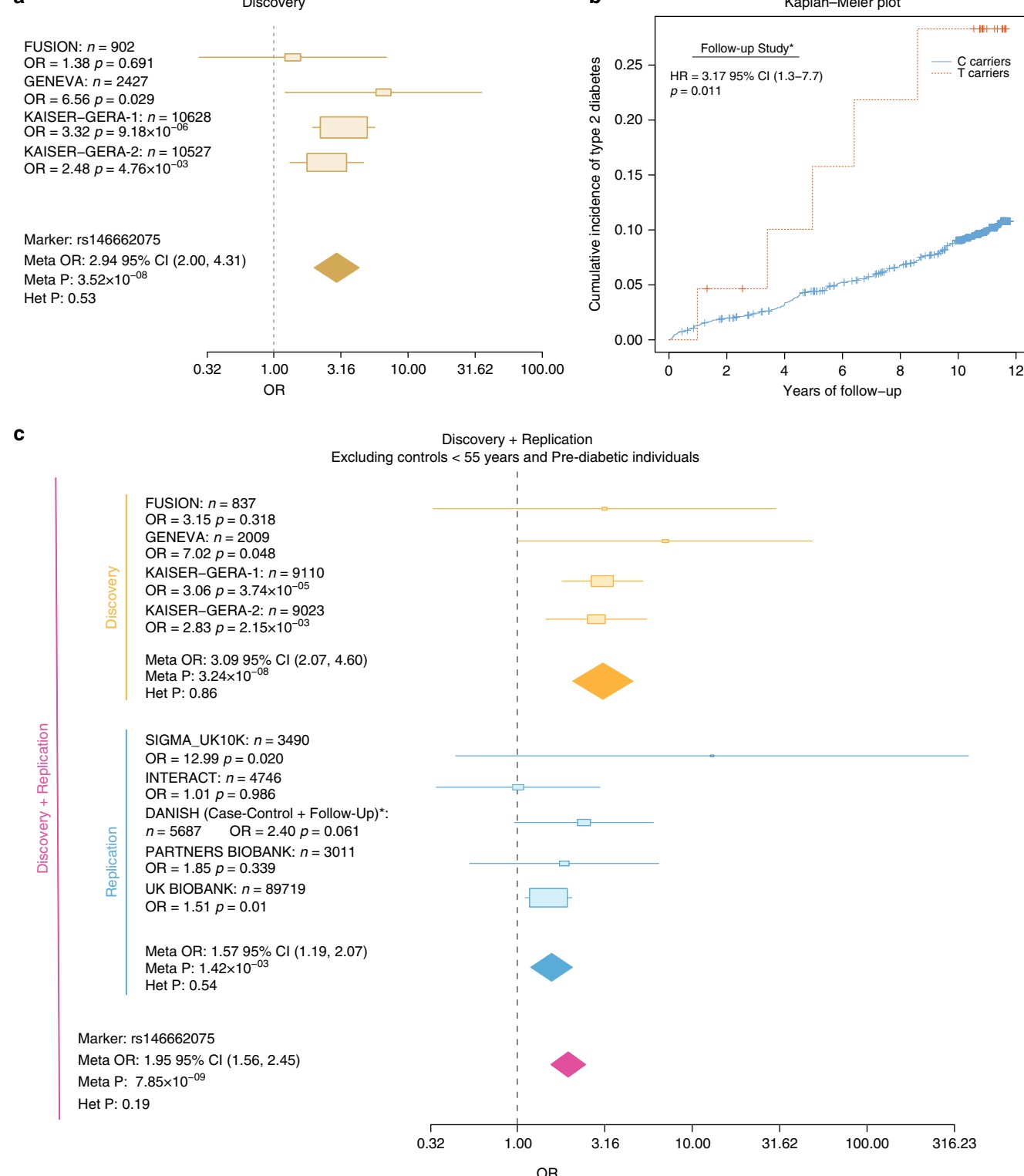

**Fig. 3** Discovery and replication of rs14666075 association signal. **a** Forest plot of the discovery of rs146662075 variant. Cohort-specific odds ratios are denoted by boxes proportional to the size of the cohort and 95% CI error bars. The combined OR estimated for all the data sets is represented by a diamond, where the diamond width corresponds to 95% CI bounds. The p-value for the meta-analysis (Meta P) and for the heterogeneity (Het P) of odds ratio is shown. **b** Kaplan–Meier plot showing the cumulative incidence of T2D for a 11 years follow-up. The red line represents the T carriers and in light blue, C carriers are represented ($n = 1,652$, cases $= 158$). **c** Forest plot after excluding controls younger than 55 years, OGTT $>7.8$ mmol $l^{-1}$, and controls with family history of T2D in both the discovery and replication cohorts when available

genetic data[6, 54]. For example, we provided for the first time a comprehensive coverage of structural variants, which point to previously unobserved candidate causal variants in known and novel loci, as well as a comprehensive coverage of the X chromosome through sequence-based imputation.

This study also highlights the importance of a strict classification of both cases and controls, in order to identify rare variants associated with disease. Our initial discovery of the Xq23 locus was only replicated when the control group was restricted to T2D-free individuals who were older than 55 years (average age

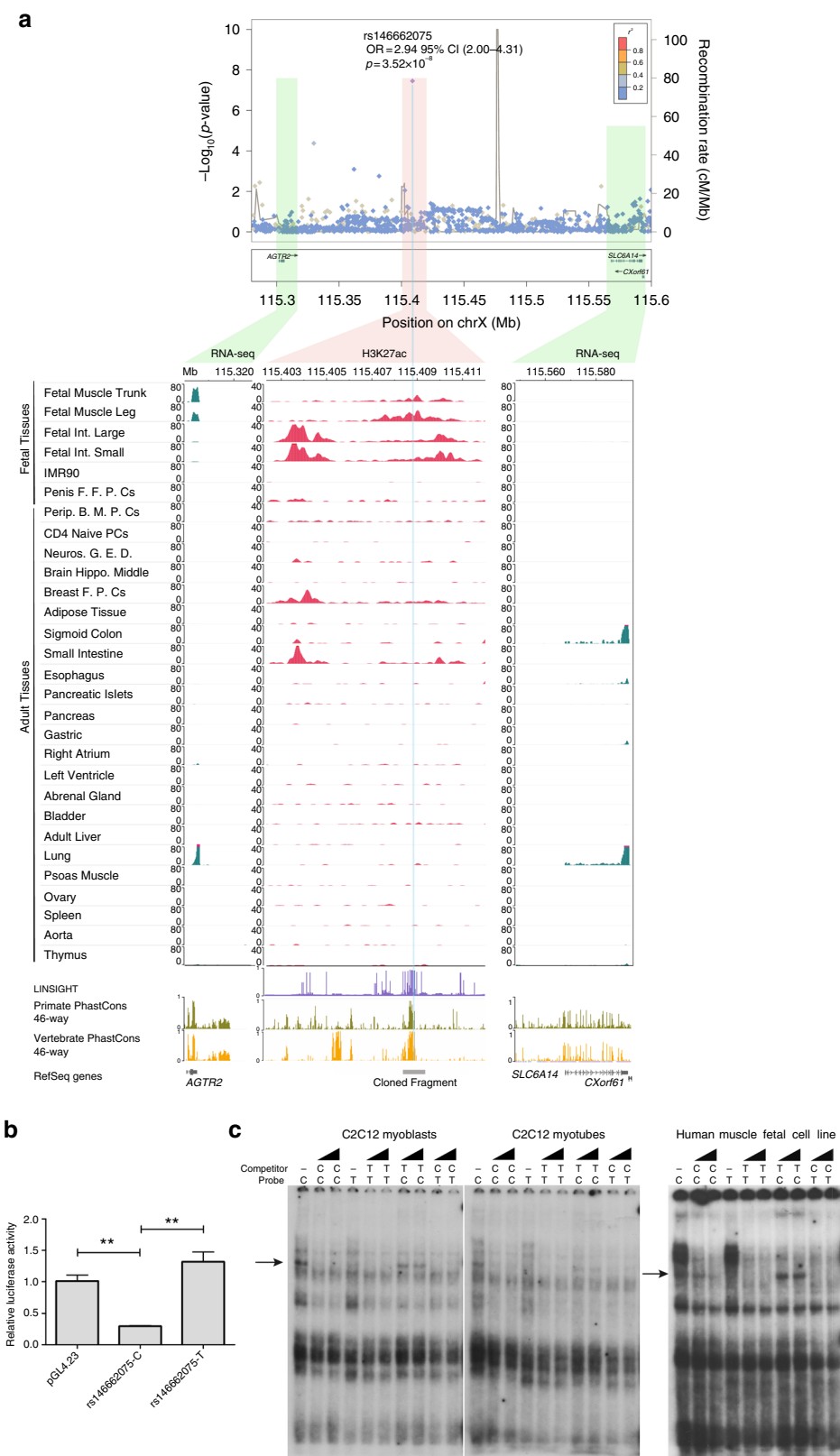

at the onset of T2D), had normal glucose tolerance, and no family history of T2D. This is in line with previous results obtained for a T2D population-specific variant found in Inuit within the *TBC1D4* gene, which was only significant when using OGTT as criteria for classifying cases and controls, but not when using HbA1c[52]. Our observation that 30% of the rs146662075 risk allele carriers developed T2D over 11 years of follow-up, compared to 10% of noncarriers, further supports the association of this variant and suggests that an early identification of these subjects through genotyping may be useful to tailor pharmacological or lifestyle intervention to prevent or delay the onset of T2D.

Using binding and gene-reporter analyses, we demonstrated a functional role of this variant and proposed a possible mechanism behind the pathophysiology of T2D in T risk allele carriers, in which this rare variant could favor a gain of function of *AGTR2*, previously associated with insulin resistance[48]. *AGTR2* appears, therefore, as a potential therapeutic target for this disease, which would be in line with previous studies showing that the blockade of the renin–angiotensin system in mice[55] and in humans[56] prevents the onset of T2D, and restores normoglycemia[57, 58].

Overall, beyond our significant contribution toward expanding the number of genetic associations with T2D, our study also highlights the potential of the reanalysis of public data, as a complement to large studies that use newly generated data. This study informs the open debate in favor of data sharing and democratization initiatives[4, 59], for investigating the genetics and pathophysiology of complex diseases, which may lead to new preventive and therapeutic applications.

## Methods

**Quality filtering for imputed variants**. In order to assess genotype imputation quality and to determine an accurate post-imputation quality filter, we made use of the Wellcome Trust Case Control Consortium (WTCCC)[40] data available through the European Genotype Archive (EGA, https://www.ebi.ac.uk/ega/studies/EGAS00000000028). The genotyping data and the subjects included in the following tests were filtered according to the guidelines provided by the WTCCC, whose criteria of exclusion are in line with standard quality filters for GWAS[60]. We used the 1958 British Birth cohort (~3,000 samples, 58C) that was genotyped by Affymetrix v6.0 and Illumina 1.2M chips. After applying the quality-filtering criteria, 2,706 and 2,699 subjects from the Affymetrix and Illumina data, respectively, were available for the 58C samples, leaving an intersection of 2,509 individuals genotyped by both platforms. After variant quality filtering and excluding all the variants with minor allele frequency (MAF) below 0.01, 717,556, and 892,516 variants remained for 58C Affymetrix and Illumina platforms, respectively.

We used a two-step genotype imputation approach based on prephasing the study genotypes into full haplotypes with SHAPEIT2[61] to ameliorate the computational burden required for genotype imputation through IMPUTE2[62]. We used the GTOOL software (http://www.well.ox.ac.uk/~cfreeman/software/gwas/gtool.html, version 0.7.5) to homogenize strand annotation by merging the imputed results obtained from each set of genotyped data. To ensure that there were no strand orientation issues, we excluded all C/G and A/T SNPs. To perform genotype imputation, we used two sequence-based reference panels: the 1000G Phase1 (June 2014) release[7] and the UK10K[2].

We evaluated genotype imputation for each reference panel considering 2,509 58C individuals that were genotyped by both independent genotyping platforms. Four scenarios were considered: (a) fraction of variants originally genotyped (GT) by both Illumina (IL) and Affymetrix (Affy) platforms (both GT), (b) variants genotyped by Affy, but not present in IL array (Affy GT), (c) variants genotyped by IL, but not present in the Affy array (IL GT), and (d) variants not typed in IL nor in the Affy arrays, and therefore, imputed from IL and Affy data sets (d). This last scenario comprised the largest fraction of variants.

As the individuals typed (and imputed) using Affy and IL SNPs as backbones were the same, we expected no statistical differences when comparing the allele and genotype frequencies with any of the variants. The quality of the imputed variants was evaluated using the allelic dosage $R^2$ correlation coefficient, between the genotype dosages estimated when imputing using Affy or IL as the backbone. The Affy GT and IL GT SNPs were used to evaluate the correspondence between the allelic dosage $R^2$ scores and the IMPUTE2 info scores for the imputed genotypes. The linear model, between the allelic dosage $R^2$ and the IMPUTE2-info, was used to set an info score threshold of 0.7, which corresponds to an allelic dosage $R^2$ of 0.5. The correlation between $R^2$ and info score was uniform across all reference panels and platforms.

**The 70KforT2D resource**. We collected genetic individual-level data for T2D case/control studies from five independent datasets, Gene Environment-Association Studies initiative [GENEVA], Wellcome Trust Case Control Consortium [WTCCC], Finland–United States Investigation of NIDDM Genetics [FUSION], Resource for Genetic Epidemiology Research on Aging [GERA], and the Northwestern NUgene project [NuGENE] publicly available in the dbGaP (http://www.ncbi.nlm.nih.gov/gap) and EGA (https://www.ebi.ac.uk/ega/home) public repositories, comprising a total of 13,201 cases and 59,656 controls (for the description of each cohort, see Supplementary Note 1 and Supplementary Data 1).

Each dataset was independently harmonized and quality controlled with a three-step protocol, including two stages of SNP removal and an intermediate stage of sample exclusion. The exclusion criteria for variants were (i) missing call rate ≥ 0.05, (ii) significant deviation from Hardy–Weinberg equilibrium (HWE) $p ≤ 1 × 10^{-6}$ for controls and $p ≤ 1 × 10^{-20}$ for the entire cohort, (iii) significant differences in the proportion of missingness between cases and controls $p ≤ 1 × 10^{-6}$, and (iv) MAF < 0.01 (for the GERA cohort, we considered a MAF of 0.001). The exclusion criteria for samples were i) gender discordance between the reported and genetically predicted sex, ii) subject relatedness (pairs with $\pi ≥ 0.125$ from which we removed the individual with the highest proportion of missingness), iii) missing call rates per sample ≥ 0.02, and iv) population structure showing more than four standard deviations within the distribution of the study population according to the first four principal components.

We performed genotype imputation independently for each cohort by prephasing the genotypes to whole haplotypes with SHAPEIT2 and then, we performed genotype imputation with IMPUTE2. We tested for association with additive derived logistic regression using SNPTEST, seven derived principal components sex, age, and body-mass index (BMI), except for WTCCC, for which age and BMI were not available (Supplementary Data 1). To maximize power and accuracy, we combined the association results from 1000G Phase1 integrated haplotypes (June, 2014)[7] and UK10K (http://www.uk10k.org/) reference panels by choosing for each variant, the reference panel that provided the best IMPUTE2 info score. For 1000G-based genotype imputation in chromosome X (chrX), we used the "v3.macGT1" release (August, 2012). For chrX, we restricted the analysis to non-pseudoautosomal (non-PAR) regions and stratified the association analysis by sex to account for hemizygosity for males, while for females, we followed an autosomal model. Also, we did not apply HWE filtering in the X chromosome variants. Finally, for the GERA cohort due to the large computational burden that comprises the whole genotype imputation process in such a large sample size, we randomly split this cohort into two homogeneous subsets of ~30,000 individuals each, in order to minimize the memory requirements.

We included variants with IMPUTE2 info score ≥ 0.7, MAF ≥ 0.001, and for autosomal variants, HWE controls $p > 1 × 10^{-6}$. Further details about genotype imputation and covariate information used in association testing are summarized in Supplementary Data 1.

**70KforT2D and inclusion of previous summary statistics data**. We meta-analyzed the different sets from the 70KforT2D data set with METAL[63], using the inverse variance-weighted fixed effect model. We included variants with $I^2$ heterogeneity < 75. This filter was not applied to the final X chromosome data set, after meta-analyzing the results from males and females separately (which were already filtered by $I^2 < 75$).

For the meta-analysis with the DIAGRAM trans-ethnic study[8], we excluded from the whole 70KforT2D datasets those cohorts that overlapped with the DIAGRAM data. Therefore, we meta-analyzed the GERA and NuGENE cohorts (7,522 cases and 50,446 controls) from the 70KforT2D analysis with the trans-ethnic summary statistics results. As standard errors were not provided for the

**Fig. 4** Functional characterization of rs146662075 association signal. **a** Signal plot for X chromosome region surrounding rs146662075. Each point represents a variant, with its p-value (on a −log10 scale, y axis) derived from the meta-analysis results from association testing in males. The x axis represents the genomic position (hg19). Below, representation of H3K27ac and RNA-seq in a subset of cell types is shown. The association between RNA-seq signals and H3K27ac marks suggests that *AGTR2* is the most likely regulated gene by the enhancer that harbors rs146662075. **b** The presence of the common allelic variant rs146662075-C reduces enhancer activity in luciferase assays performed in a mouse myoblast cell line. **c** Electrophoretic mobility shift assay in C2C12 myoblast cell lines, C2C12-differentiated myotubes, and human fetal myoblasts showed allele-specific binding of a ubiquitous nuclear complex. The arrows indicate the allele-specific binding event. Competition was carried out using 50- and 100-fold excess of the corresponding unlabeled probe

DIAGRAM trans-ethnic meta-analysis, we performed a sample size based meta-analysis, which converts the direction of the effect and the p-value into a Z-score. In addition, we also performed an inverse variance-weighted fixed effect meta-analysis to estimate the final effect sizes. This approach required the estimation of the beta and standard errors from the summary statistics (p-value and odds ratio).

For the meta-analysis of coding low-frequency variants with the Type 2 Diabetes Knowledge Portal (T2D Portal)[6], we included from the 70KforT2D data set the NuGENE and GERA cohorts (7,522 cases and 50,446 controls), to avoid overlapping samples. Like in the previous scenario, standard errors were not provided for the T2D Portal data and we used a sample size based meta-analysis with METAL. However, to estimate the effect sizes, we also calculated the standard errors from the p-values and odds ratios, and we performed an inverse variance-weighted fixed effect meta-analysis.

See further details about the cohorts in Supplementary Note 1.

**Pathway and enrichment analysis**. Summary statistics that resulted from the 70KforT2D meta-analysis were analyzed by Data-driven Expression-Prioritized Integration for Complex Traits (DEPICT)[9] to prioritize likely causal genes, to highlight enriched pathways, and to identify the most relevant tissues/cell types; DEPICT relies on publicly available gene sets (including molecular pathways) and leverages gene expression data from 77,840 gene expression arrays, to perform gene prioritization and gene-set enrichment based on predicted gene function and the so-called reconstituted gene sets. A reconstituted gene set contains a membership probability for each gene and conversely, each gene is functionally characterized by its membership probabilities across 14,461 reconstituted gene sets. As an input to DEPICT, we used all summary statistics from autosomal variants with $p < 1 \times 10^{-5}$ in the 70KforT2D meta-analysis. We used an updated version of DEPICT, which handled 1000G Phase1-integrated haplotypes (June 2014, www.broadinstitute.org/depict). DEPICT was run using 3,412 associated SNPs ($p < 1 \times 10^{-5}$), from which we identified independent SNPs using PLINK and the following parameters: --clump-p1 5e-8, --clump-p2 1e-5, --clump-r2 0.6, and --clump-kb 250. We used LD $r^2 > 0.5$ distance to define locus limits yielding 70 autosomal loci comprising 119 genes (note that this is not the same locus definition that we used elsewhere in the text). We ran DEPICT with default settings, i.e., using 500 permutations to adjust for bias and 50 replications to estimate false discovery rate (FDR). We used normalized expression data from 77,840 Affymetrix microarrays to reconstitute gene sets[9]. The resulting 14,461 reconstituted gene sets were tested for enrichment analysis. A total of 209 tissue or cell types expression data assembled from 37,427 Affymetrix U133 Plus 2.0 Array samples were used for enrichment in tissue/cell-type expression. DEPICT identified 103 reconstituted gene sets significantly enriched (FDR < 5%) for genes found among the 70 loci associated to T2D. We did not consider reconstituted sets in which genes of the original gene set were not nominally enriched (Wilcoxon rank-sum test), as these are expected to be enriched in the reconstituted gene set by design. The lack of enrichment makes the interpretation of the reconstituted gene set challenging because the label of the reconstituted gene set will not be accurate. Hence, the following reconstituted gene sets were removed from the results (Wilcoxon rank sum and P-values in parentheses): MP:0004247 gene set ($p = 0.73$), GO:0070491 gene set ($p = 0.14$), MP:0004086 gene set ($p = 0.17$), MP:0005491 gene set ($p = 0.54$), GO:0005159 gene set ($p = 0.04$), MP:0005666 gene set ($p = 0.05$), ENSG00000128641 gene set ($p = 0.02$), MP:0006344 gene set ($p = 0.42$), MP:0004188 gene set ($p = 0.22$), MP:0002189 gene set ($p = 0.02$), MP:0000003 gene set ($p = 0.08$), ENSG00000116604 gene set ($p = 0.13$), GO:0005158 gene set ($p = 0.07$), and MP:0001715 gene set ($p = 0.01$). After applying the filters described above, there were 89 significantly enriched reconstituted gene sets. We used the affinity propagation tool to cluster related reconstituted gene sets (network diagram script available from https://github.com/perslab/DEPICT).

We also used the VSE R package to compute the enrichment or depletion of genetic variants comprised in the 57 credible sets listed in Supplementary Data 5 across regulatory genomic annotations, as described in[64]. Each GWAS lead variant from the final meta-analysis was considered as a tag SNP and variants from the corresponding 99% credible set (Supplementary Data 5) in LD with the tag SNP ($R^2 \geq 0.4$), as a cluster or associated variant set (AVS). In order to account for the size and structure of the AVS, a null distribution was built based on random permutations of the AVS. Each permuted variant set was matched to the original AVS, cluster by cluster using HapMap data by size and structure. This Matched Random Variant Set (MRVS) was calculated using 500 permutations. Significant enrichments or depletions were considered when the Bonferroni-adjusted p-value was < 0.01. Human islet regulatory elements (C1–C5) were obtained from[10].

**Definition of 99% credible sets of GWAS-significant loci**. For each genome-wide significant region locus, we identified the fraction of variants that have, in aggregate, 99% probability of containing the causal T2D-associated variant. By using our 70KforT2D meta-analysis based on imputed data (NuGENE, GERA, FUSION, GENEVA, and WTCCC data sets, comprising 12,231 cases and 57,196 controls), we defined the 99% credible set of variants for each locus with a Bayesian refinement approach[11] (we considered variants with an $R^2 > 0.1$ with their respective leading SNP).

Credible sets of variants are analogous to confidence intervals as we assume that the credible set for each associated region contains, with 99% probability, the true

causal SNP if this has been genotyped or imputed. The credible set construction provides, for each variant placed within a certain associated locus, a posterior probability of being the causal one[11]. We estimated the approximate Bayes' factor (ABF) for each variant as

$$\text{ABF} = \sqrt{1-r}\, e^{(rz^2/2)},$$

where

$$r = \frac{0.04}{(\text{SE}^2 + 0.04)},$$

$$z = \frac{\beta}{\text{SE}}.$$

The $\beta$ and the SE are the estimated effect size and the corresponding standard error resulting from testing for association under a logistic regression model. The posterior probability for each variant was obtained as

$$\text{Posterior Probability}_i = \frac{\text{ABF}_i}{T},$$

where $ABF_i$ corresponds to the approximate Bayes' factor for the marker $i$ and $T$ represents the sum of all the $ABF$ values from the candidate variants enclosed in the interval being evaluated. This calculation assumes that the prior of the $\beta$ corresponds to a Gaussian with mean 0 and variance 0.04, which is also the same prior commonly employed by SNPTEST, the program being used for calculating single-variant associations.

Finally, we ranked variants according to the $ABF$ (in decreasing order) and from this ordered list, we calculated the cumulative posterior probability. We included variants in the 99% credible set of each region until the SNP that pushed the cumulative posterior probability of association over 0.99.

The 99% credible sets of variants for each of the 57 GWAS-significant regions are summarized in Supplementary Data 5.

**Characterization of indels**. We examined whether indels from the 99% credible sets were present or absent in the 1000G Phase1 or UK10K reference panels, and also checked whether they were present or not in the 1000G Phase3 reference panel. All the information has been summarized in Supplementary Data 6. We also visually inspected the aligned BAM files of the most relevant indels from both projects to discard that they could be alignment artifacts.

**Functional annotation of the 99% credible set variants**. To determine the effect of 99% credible set variants on genes, transcripts, and protein sequence, we used the variant effect predictor (VEP, GRCh37.p13 assembly)[13]. The VEP application determines the effect of variants (SNPs, insertions, deletions, CNVs, or structural variants) on genes, transcripts, proteins, and regulatory regions. We used as input the coordinates of variants within 99% credible sets and the corresponding alleles, to find out the affected genes and RefSeq transcripts and the consequence on the protein sequence by using the GRCh37.p13 assembly. We also manually checked all these annotations with the Exome Aggregation Consortium data set (ExAC, http://exac.broadinstitute.org) and the most updated VEP server based on the GRCh38.p7 assembly. All these annotations are provided in Supplementary Data 7.

We used combined annotation-dependent depletion (CADD) scoring function to prioritize functional, deleterious, and disease causal variants. We obtained the scaled C-score (PHRED-like scaled C-score ranking each variant with respect to all possible substitutions of the human genome) metric for each 99% credible set variant, as it highly ranks causal variants within individual genome sequences[14] (Supplementary Data 8). We also used the LINSIGHT score to prioritize functional variants, which measures the probability of negative selection on noncoding sites by combining a generalized linear model for functional genomic data with a probabilistic model of molecular evolution[15]. For each credible set variant, we retrieved the precomputed LINSIGHT score at that particular nucleotide site, as well as the mean LINSIGHT precomputed score for a region of 20 bp centered on each credible set variant, respectively (https://github.com/CshlSiepelLab/LINSIGHT). These metrics are summarized in Supplementary Data 9.

In order to prioritize functional regulatory variants, we used the V6 release from the GTEx data that provides gene-level expression quantifications and eQTL results based on the annotation with GENCODE v19. This release included 450 genotyped donors, 8,555 RNA-seq samples across 51 tissues, and two cell lines, which led to the identification of eQTLs across 44 tissues[16]. Moreover, RNA-seq data from human pancreatic islets from 89 deceased donors cataloged as eQTLs and exon use (sQTL) were also integrated with the GWAS data to prioritize candidate regulatory variants[17] but in pancreatic islets, which is a target tissue for T2D. Both analyses are summarized in Supplementary Data 10 and Supplementary Data 11, respectively.

**Conditional analysis**. To confirm the independence between novel loci and previously known T2D signals, we performed reciprocal conditional analyses (Supplementary Data 5, Supplementary Data 12, Supplementary Data 13, and Supplementary Data 14). We included the conditioning SNP as a covariate in the

logistic regression model, assuming that every residual signal that arises corresponds to a secondary signal independent from this conditioning SNP. We applied this method to the *EHMT2* locus (less than 1Mb away from the *HLA* where T2D and T1D signals have been identified), to confirm that this association was independent of previously reported T2D signals and also to discard that this association is also driven by possible contamination of T1D diagnosed as T2D cases. We conditioned on the top variant identified in this study and the top variant from the 99% credible set analysis, but also on the top variants previously described for T2D and T1D[8, 38–40]. For this purpose, we used the full 70KforT2D resource (NuGENE, GERA, FUSION, GENEVA, and WTCCC cohorts imputed with 1000G and UK10K reference panels). Finally, all the results were meta-analyzed as explained in previous sections. These analyses are provided in Supplementary Data 13. This approach was also applied to confirm that the novel *CAMKK2* signal at rs3794205 is independent of known T2D signals at the *HNF1A* locus (rs1169288, rs1800574, and chr12:121440833:D)[54], which is summarized in Supplementary Data 12. Moreover, this approach confirmed known secondary signals in the 9p21 locus[65] which allowed us to build 99% credible sets based on the results from the conditional analyses (included in Supplementary Data 5), and allowed us to identify the most likely causal variant for the *DUSP9* locus (Supplementary Data 14).

**Replication of the rare variant association at Xq23.** To replicate the association of the rs146662075 variant, we performed genotype imputation with the UK10K reference panel in four independent data sets: the InterAct case-cohort study[43], the Slim Initiative in Genomic Medicine for the Americas (SIGMA) consortium GWAS data set[6], the Partners HealthCare Biobank (Partners Biobank) data set[44], and the UK Biobank cohort[45]. Phasing was performed with SHAPEIT2 and the IMPUTE2 software was used for genotype imputation.

The current UK Biobank data release did not contain imputed data for the X chromosome, for which phasing and imputation had to be analyzed in-house. The data release used comprises X chromosome QCed genotypes of 488,377 participants, which were assayed using two arrays sharing 95% of marker content (Applied Biosystems$^{TM}$ UK BiLEVE Axiom$^{TM}$ Array and the Applied Biosystems$^{TM}$ UK Biobank Axiom$^{TM}$ Array). We included samples and markers that were used as input for phasing by UK Biobank investigators. At the sample level, we also excluded women, individuals with missing call rate > 5% or showing gender discordance between the reported and the genetically predicted sex. At the variant level, we excluded markers with MAF < 0.1% and with missing call rate > 5%. The final set of 16,463 X chromosome markers and 222,725 male individuals was split into six subsets due to the huge computational burden that would require phasing into whole haplotypes the entire data set. We also excluded indels, variants with MAF < 1%, and variants showing deviation of Hardy–Weinberg equilibrium with $p < 1 \times 10^{-20}$ before the imputation step. In addition, from those pairs of relatives reported to be third degree or higher according to UK Biobank, we excluded from each pair the individual with the lowest call rate. We then tested the rs146662075 variant for association with type 2 diabetes using SNPTEST v2.5.1 and the threshold method. To avoid contamination from other types of diabetes mellitus, we excluded from the entire sample data set, individuals with ICD10 codes falling in any of these categories: E10 (insulin-dependent diabetes mellitus), E13 (other specified diabetes mellitus), and E14 (unspecified diabetes mellitus). Then, we designated as T2D cases those individuals with E11 (non-insulin-dependent diabetes mellitus) ICD10 codes, and the rest as controls. Moreover, we only kept as control subjects those individuals without reported family history of diabetes mellitus and older than 55 years, which is the average age at the onset of T2D.

We also genotyped de novo the rs146662075 variant with KASPar SNP genotyping system (LGC Genomics, Hoddeson, UK) in the Danish cohort, which comprises data from five sample sets (Supplementary Note 2 also for the genotyping and QC analysis for this variant).

We used Cox-proportional hazard regression models to assess the association of the variant with the risk of incident T2D in 1,652 nondiabetic male subjects genotyped in the Inter99 cohort (part of the Danish cohort) that were followed for 11 years on average. The follow-up analysis was restricted to male individuals younger than 45 years who were 56 years old after 11 years of follow-up. Individuals with self-reported diabetes at the baseline examination and individuals present in the Danish National diabetes registry before the baseline examination were also excluded. To include the follow-up study as a part of the replication cohorts, we used a meta-analysis method that accounts for overlapping samples (MAOS)[66], as we had to control for the sample overlap between the follow-up and the case-control study from the Danish samples.

See Supplementary Note 2 for a larger description of each of the five replication cohorts and how they have been processed.

We meta-analyzed the association results from these five replication data sets with the 70KforT2D data sets. In the final meta-analysis, we excluded whenever it was possible (a) controls younger than 55 years and (b) with OGTT > 7.8 mmol l$^{-1}$ or with family history of T2D.

**In silico functional characterization of rs146662075.** This variant is located in an intergenic region, flanked by *AGTR2* and *SLC6A14* genes, and within several DNase I hypersensitive sites. We searched for regulatory marks (i.e., H3K4me1 and H3K27ac marks) through the HaploReg web server (http://archive.broadinstitute.

org/mammals/haploreg/haploreg.php), in order to assess which type of regulatory element was associated with the rs146662075 variant.

To further evaluate the putative regulatory role of rs146662075, we used the WashU EpiGenome Browser (http://epigenomegateway.wustl.edu/browser/, last access on June 2016). We used the following public data hubs: (1) the reference human epigenomes from the Roadmap Epigenomics Consortium track hubs and (2) the Roadmap Epigenomics Integrative Analysis Hub. These data were released by the NIH Roadmap Epigenomics Mapping Consortium[51]. RNA-seq data were used to evaluate whether gene expression of any of the closest genes (*AGTR2* and *SLC6A14* genes, fixed scale at 80 RPKM) correlated with the presence of the H3K27ac enhancer marks (a more strict mark for active enhancers in contrast with H3K4me1[67], which were highlighted by the HaploReg search) at the rs146662075 location. For visualizing the H3K27ac marks around rs146662075, we focused on a region of 8 kb and we used a fixed scale at 40 −log$_{10}$ Poisson *p*-value of the counts relative to the expected background count ($\lambda_{local}$).

The NIH Roadmap Epigenomics Consortium data from standardized epigenomes also allowed us to further interrogate which target gene within the same topologically associating domain (TAD) was more likely to be regulated by this rs146662075 enhancer. We used H3K27ac narrow peaks from 59 tissues called using MACSv2 with a *p*-value threshold of 0.01 from 98 consolidated epigenomes to seek for enhancer marks in a given tissue (the presence of H3K27ac peak). To assess gene expression for any of the putative target genes within TAD, we used the RPKM expression matrix for 57 consolidated epigenomes (http://egg2.wustl.edu/roadmap/data/byDataType/rna/) and gene expression quantifications for fetal muscle leg, fetal muscle trunk, and fetal stomach provided by ENCODE (https://www.encodeproject.org/). With this, we were able to test for each of the genes, the association between gene expression and enhancer activity in 31 tissues with a Fisher's exact test.

**Allele-specific enhancer activity at rs146662075.** The mouse C2C12 cell line (ATCC CRL-1772) was grown in DMEM medium supplemented with 10% FBS and was induced to differentiate in DMEM with 10% horse serum for 4 days.

The human fetal myoblast cell line was established by Prof. Giulio Cossu (Institute of Inflammation and Repair, University of Manchester)[68]. The authors played no role in the procurement of the tissue. Cells were cultured in DMEM medium supplemented with 10% fetal calf serum and was induced to differentiate in DMEM with 2% horse serum for 4 days.

To perform an electrophoretic mobility shift assay, nuclear extracts from mouse myoblast C2C12 cells and the human myoblast cell line (ATCC CRL-1772) were obtained as described before[69]. Double-stranded oligonucleotides containing either the common or rare variants of rs146662075 were labeled using dCTP [α-32P] (Perkin Elmer). Oligonucleotide sequences are as follows (SNP location is underlined): probe-C-F: 5′-gatcTTTGAACACcGAGGGGAAAAT-3′ and R:5′-gatcATTTTCCCCTC gGTGTTCAAA-3′ and probe-T-F: 5′- gatcTTTGAACACtGAGGGGAAAAT-3′ and R: 5′-gatcATTTTCCCCTCaGTGTTCAAA-3′. Assay specificity was assessed by preincubation of nuclear extracts with 50- and 100-fold excess of unlabeled wild-type or mutant probes, followed by electrophoresis on a 5% nondenaturing polyacrylamide gel. Findings were confirmed by repeating binding assays on separate days.

For evaluating if the activity of the rs146662075 enhancer was allele specific, we performed a luciferase assay. A region of 969 bp surrounding rs146662075 was amplified from human genomic DNA using F: 5′-GCTAGCATATGGAGGTGATTTGT-3′ and R: 5′-GGCACTTCCTTCTCTGGTAGA-3′ oligonucleotides and cloned into pENTR/D-TOPO (Invitrogen). Allelic variant rs146662075T was introduced by site-directed mutagenesis using the following primers: F: 5′-CCTTTTTTTTACTTTGAACACTGAGGGGAAAATCATGCTTGGC-3′ and R: 5′-GCCAAGCATGATTTTCCCCTCAGTGTTCAAAGTAAAAAAAGG-3′. Enhancer sequences were shuttled into pGL4.23[luc2/minP] vector (Promega) adapted for Gateway cloning (pGL4.23-GW, **2**) using Gateway LR Clonase II Enzyme mix (Invitrogen). Correct cloning was confirmed both by Sanger sequencing and restriction digestion.

C2C12 (ATCC CRL-1772) and 293T (ATCC CRL-3216) cells were transfected in quadruplicates with 500 ng of pGL4.23-GW enhancer containing vectors and 0.2 ng of Renilla normalizer plasmid. Transfections were carried out in 24-well plates using Lipofectamine 2000 and Opti-MEM (Thermo Fisher Scientific) following the manufacturer's instructions. Luciferase activity was measured 48 h after transfection using Dual-Luciferase Reporter Assay System (Promega). Firefly luciferase activity was normalized to Renilla luciferase activity, and the results were expressed as a normalized ratio to the empty pGL4.23[luc2/minP] vector backbone. Experiments were repeated three times. Statistical significance was evaluated through a Student's *t*-test.

**Data availability**. The association results are available at the Type 2 Diabetes Knowledge portal (www.type2diabetesgenetics.org/) and the complete summary statistics are available for download at http://cg.bsc.es/70kfort2d/

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

## Acknowledgements

This work has been sponsored by the grant SEV-2011-00067 of Severo Ochoa Program, awarded by the Spanish Government. This work was supported by an EFSD/Lilly research fellowship. Josep M. Mercader was supported by Sara Borrell Fellowship from the Instituto Carlos III and Beatriu de Pinós fellowship from the Agency for Management of University and Research Grants (AGAUR). Sílvia Bonàs was FI-DGR Fellowship from FI-DGR 2013 from Agència de Gestió d'Ajuts Universitaris i de Recerca (AGAUR, Generalitat de Catalunya). This study makes use of data generated by the WTCCC. A full list of the investigators who contributed to the generation of the data is available from www.wtccc.org.uk. Funding for the project was provided by the Wellcome Trust under award 076113. This study also makes use of data generated by the UK10K Consortium, derived from samples from UK10K COHORT IMPUTATION (EGAS00001000713). A full list of the investigators who contributed to the generation of the data is available in www.UK10K.org. Funding for UK10K was provided by the Wellcome Trust under award WT091310. We acknowledge PRACE for awarding us to access MareNostrum supercomputer, based in Spain at Barcelona. The technical support group, particularly Pablo Ródenas and Jorge Rodríguez, from the Barcelona Supercomputing Center is gratefully acknowledged. This project has received funding from the European Union's Horizon 2020 research and innovation program under grant agreement No 667191. Mercè Planas-Fèlix is funded by the Obra Social Fundación la Caixa fellowship under the Severo Ochoa 2013 program. Work from Irene Miguel-Escalada, Ignasi Moran, Goutham Atla, and Jorge Ferrer was supported by the National Institute for Health Research (NIHR) Imperial Biomedical Research Centre, the Wellcome Trust (WT101033), Ministerio de Economía y Competitividad (BFU2014-54284-R) and Horizon 2020 (667191). Irene Miguel-Escalada has received funding from the European Union's Horizon 2020 research and innovation program under the Marie Sklodowska–Curie grant agreement No 658145. We acknowledge Prof. Giulio Cossu (Institute of Inflammation and Repair, University of Manchester) for providing the muscle myoblast cell line. We also acknowledge the InterAct and SIGMA Type 2 Diabetes Consortia for access to the data to replicate the rs146662075 variant. A full list of the investigators of the SIGMA Type 2 Diabetes and the InterAct consortia is provided in Supplementary Notes 3 and 4. The Novo Nordisk Foundation Center for Basic Metabolic Research is an independent research center at the University of Copenhagen partially funded by an unrestricted donation from the Novo Nordisk Foundation (www.metabol.ku.dk). This research has been conducted using the UK Biobank Resource (application number 16803). We also acknowledge Bianca C. Porneala, MS for his technical assistance in the collection and curation of the genotype and phenotype data from Partners Biobank. We also thank Marcin von Grotthuss for their support for uploading the summary statistics data to the Type 2 Diabetes Genetic Portal (AMP-T2D portal). Finally, we thank all the Computational Genomics group at the BSC for their helpful discussions and valuable comments on the manuscript.

## Author contributions

S.B-G., J.M.M., and D.T. conceived, planned, and performed the main analyses. S.B-G., J.M.M., and D.T. wrote the manuscript. M.G-M., F.S., P.C-S., M.P., C.D., and R.M.B. developed a framework for large-scale imputation analyses. E.R-F., P.T., and T.H.P. performed pathway analysis. I.M-E. performed the enrichment analysis. M.P-F. and S.G. performed structural variant analyses. N.G., J.R-G., J.M., E.A.A., M.U., A.L., V.K., J.F., T.J., A.L., M.E.J., D.R.W., C.C., I.B., E.V.A., R.A.S., J.L., C.L., N.J.W., O.P., J.C.F., and T.H. contributed with additional data and analyses. G.A., I.M., and C.C.M. performed additional bioinformatics analyses. D.S. and A.Z. contributed muscle cell lines. I.M-E. and J.F. performed luciferase and electrophoretic mobility shift assays. J.M.M. and D.T. designed and supervised the study. All authors reviewed and approved the final manuscript.

## Additional information

**Competing interests:** The authors declare no competing financial interests.

Sílvia Bonàs-Guarch[1], Marta Guindo-Martínez[1], Irene Miguel-Escalada[2,3,4], Niels Grarup[5], David Sebastian[3,6,7], Elias Rodriguez-Fos[1], Friman Sánchez[1,8], Mercè Planas-Fèlix[1], Paula Cortes-Sánchez[1], Santi González[1], Pascal Timshel[5,9], Tune H. Pers[5,9,10,11], Claire C. Morgan[4], Ignasi Moran[4], Goutham Atla[2,3,4], Juan R. González[12,13,14], Montserrat Puiggros[1], Jonathan Marti[8], Ehm A. Andersson[5], Carlos Díaz[8], Rosa M. Badia[8,15], Miriam Udler[16,17], Aaron Leong[17,18], Varindepal Kaur[17], Jason Flannick[16,17,19], Torben Jørgensen[20,21,22], Allan Linneberg[20,23,24], Marit E. Jørgensen[25,26], Daniel R. Witte[27,28], Cramer Christensen[29], Ivan Brandslund[30,31], Emil V. Appel[5], Robert A. Scott[32], Jian'an Luan[32],

Claudia Langenberg[32], Nicholas J. Wareham[32], Oluf Pedersen[5], Antonio Zorzano[3,6,7], Jose C Florez[16,17,33], Torben Hansen [5,34], Jorge Ferrer [2,3,4], Josep Maria Mercader [1,16,17] & David Torrents [1,35]

[1]Barcelona Supercomputing Center (BSC), Joint BSC-CRG-IRB Research Program in Computational Biology, 08034 Barcelona, Spain. [2]Genomic Programming of Beta-cells Laboratory, Institut d'Investigacions August Pi i Sunyer (IDIBAPS), 08036 Barcelona, Spain. [3]Instituto de Salud Carlos III, Centro de Investigación Biomédica en Red de Diabetes y Enfermedades Metabólicas Asociadas (CIBERDEM), 28029 Madrid, Spain. [4]Section of Epigenomics and Disease, Department of Medicine, Imperial College London, London W12 0NN, UK. [5]The Novo Nordisk Foundation Center for Basic Metabolic Research, Section for Metabolic Genetics, Faculty of Health and Medical Sciences, University of Copenhagen, 2100 Copenhagen, Denmark. [6]Institute for Research in Biomedicine (IRB Barcelona), The Barcelona Institute of Science and Technology, Baldiri Reixac 10-12, 08028 Barcelona, Spain. [7]Departament de Bioquímica i Biomedicina Molecular, Facultat de Biologia, Universitat de Barcelona, 08028 Barcelona, Spain. [8]Computer Sciences Department, Barcelona Supercomputing Center (BSC-CNS), 08034 Barcelona, Spain. [9]Department of Epidemiology Research, Statens Serum Institut, 2300 Copenhagen, Denmark. [10]Division of Endocrinology and Center for Basic and Translational Obesity Research, Boston Children's Hospital, Boston, MA 02116, USA. [11]Medical and Population Genetics Program, Broad Institute of MIT and Harvard, Cambridge, MA 02142, USA. [12]ISGlobal, Centre for Research in Environmental Epidemiology (CREAL), 08003 Barcelona, Spain. [13]CIBER Epidemiología y Salud Pública (CIBERESP), 28029 Madrid, Spain. [14]Universitat Pompeu Fabra (UPF), 08003 Barcelona, Spain. [15]Artificial Intelligence Research Institute (IIIA), Spanish Council for Scientific Research (CSIC), 28006 Madrid, Spain. [16]Programs in Metabolism and Medical and Population Genetics, Broad Institute of Harvard and MIT, Cambridge, MA 02142, USA. [17]Diabetes Unit and Center for Genomic Medicine, Massachusetts General Hospital, Boston, MA 02114, USA. [18]Division of General Internal Medicine, Massachusetts General Hospital, Boston, MA 02114, USA. [19]Department of Molecular Biology, Harvard Medical School, Boston, MA 02114, USA. [20]Research Centre for Prevention and Health, Capital Region of Denmark, DK-2600 Glostrup, Denmark. [21]Faculty of Health and Medical Sciences, University of Copenhagen, DK-2200 Copenhagen, Denmark. [22]Faculty of Medicine, University of Aalborg, DK-9220 Aalborg East, Denmark. [23]Department of Clinical Experimental Research, Rigshospitalet, Glostrup, 2100 Copenhagen, Denmark. [24]Department of Clinical Medicine, Faculty of Health and Medical Sciences, University of Copenhagen, DK-2200 Copenhagen, Denmark. [25]Steno Diabetes Center, 2820 Gentofte, Denmark. [26]National Institute of Public Health, Southern Denmark University, DK-5230 Odense M, Denmark. [27]Department of Public Health, Aarhus University, DK-8000 Aarhus C, Denmark. [28]Danish Diabetes Academy, DK-5000 Odense C, Denmark. [29]Medical department, Lillebaelt Hospital, 7100 Vejle, Denmark. [30]Department of Clinical Biochemistry, Lillebaelt Hospital, 7100 Vejle, Denmark. [31]Institute of Regional Health Research, University of Southern Denmark, DK-5230 Odense, Denmark. [32]MRC Epidemiology Unit, University of Cambridge School of Clinical Medicine, Cambridge Biomedical Campus, Cambridge CB2 0QQ, UK. [33]Department of Medicine, Harvard Medical School, Boston, MA 02115, USA. [34]Faculty of Health Sciences, University of Southern Denmark, DK-5230 Odense M, Denmark. [35]Institució Catalana de Recerca i Estudis Avançats (ICREA), 08010 Barcelona, Spain. Josep Maria Mercader and David Torrents jointly supervised this work.

