## [Peer Review File · Nature Communications]

Reviewer #1 (Remarks to the Author):

Bonas-Guarch et al have performed a re-analysis of published GWAS data (available through EGA and dbGAP databases) on T2D of European ancestry including 70,127 individuals (13,857 with T2D) using novel strategies and imputation based upon 1000G and UK10K panels. A key difference to previous meta-analysis attempts using aggregated summary statistics generated independently by different centers, while these authors performed a unified re-analysis of multiple publicly available datasets applying standardized QC methods, genotype imputation and association methods across the whole genome including the X chromosome. After harmonization and QC 70,127 subjects remained (12,931 with T2D) for analyses. The meta-analysis yielded 15,115,281 (6,845,408 common, 3,100,848 low-frequency and 5,169,025 rare) variants across cases and controls with a heterogeneity score <0.75). Using both 1001G and UK10K reference panels for imputation clearly improved number of good quality variants including indels (1,357,753). The meta-analysis strategies involved both individual-level genotypes and summary statistics and queries through the www.type2diabetesgenetics.org portal. A gene-set enrichment analysis showed enrichment of genes expressed in pancreas as well as enrichment for active regulatory enhancers. These analyses identified 57 genome-wide significant loci, 7 of which had not been previously reported. They also created a set of 99% credible variants; out of 8,348 credible variants 927 were indels and 105 genome-wide significant; one such example was previously reported intronic variants in the IGFBP2 gene. For prioritization of causal variants in a locus the VEP and CADD tools were used and interrogated with GTEX. Analysis of the X chromosome not only replicated the DUSP9 gene locus but also suggested that the original association may be driven by a 3-basepair deletion. In addition, a novel signal in males was identified at Xq23 rs 146662075. Imputation for this variant was only accurate in males. Thereafter they attempted to replicate the finding by imputation in SIGMA and INTERACT cohorts as well as de-novo-genotyping in Danish individuals. They also showed that this locus had an allele-specific effect on transactivation on binding using traditional assays, Luciferase and EMSA. Interestingly, only by selecting individuals from the extremes of the distribution of 2 hr glucose during an OGTT they were able to replicate this finding. Other novel loci included a region on 9q34.2 near the ABO gene and a locus on 10q22.1 including the NEUROG3 gene. The authors emphasized that only re-analysis of publicly available data using novel analytical approaches made it possible to generate these novel genetic discoveries.

Comments

This is an interesting study applying novel analytical tools to identify new genetic variants associated with T2D using published data. The results seem convincing and for some replicated applying interesting approaches. The functional studies on rs 146662075 using traditional assays. The paper is well written, supplement structured and messages clear. The authors should be acknowledged for Figure 1, it is very nice and informative! I would though pay more attention to the following issues.

1. The strategy to use "super-controls" based upon 2 hr glucose is really interesting but why do you restrict it to your X-chromosome SNP. Many of the cohorts studied have OGTT data and you could test whether you obtain stronger associations for your new SNPs but also some "old" established like variants in TCF7L2, MTNR1B, SLC30A8 etc. This could serve as proof of principle and replication for the strategy, otherwise we are left with the view that it is only important for your X chrom SNP.
2. The most recent paper asking the same question applying novel analytical methods is ref 14. It would be valuable to discuss similarities and dissimilarities between the current ms and ref 14.
3. It would also be valuable to test the novel program LINSIGHT (Nat Gen 49,618,2017) which is supposed to outweigh all other programs for identifying human non-coding variants, better than CADD used in this ms.

4. I did not find any comment on ethics and IRB approvals. I know that this has been a prerequisite for depositing data in data-bases as dbGap, but many of the data bases used, DIAGRAM, InterAct (not INTERACT) are not primary databases but aggregates of a number of data bases with different IRB approvals. A statement on how this has been secured would be needed.

Reviewer #2 (Remarks to the Author):

Reviewer Critique

In their study, entitled "A comprehensive reanalysis of publicly available GWAS datasets reveals an X chromosome rare regulatory variant associated with high risk for type 2 diabetes", Dr. Sílvia Bonàs-Guarch and colleagues performed reanalysis of publicly available GWAS data to gain insights into the genetics of diabetes (T2D). The authors accessed GWAS data from 70,127 subjects, and used an innovative imputation and association strategy that allowed them to replicate and fine map 50 known T2D loci, and identify seven novel associated regions: five driven by common variants in or near *LYPLAL1*, *NEUROG3*, *CAMKK2*, *ABO* and *GIP* genes; one by a low frequency variant near *EHMT2*. One of the locus was driven by a rare variant in chromosome Xq23, associated with a 2.7-fold increased risk for T2D in males, and located within an active enhancer associated with the expression of Angiotensin II Receptor type 2 gene (*AGTR2*), a known modulator of insulin sensitivity. The authors further show that the risk T allele reduces binding of a nuclear protein, resulting in increased enhancer activity in muscle cells. The authors conclude that beyond providing novel insights into the genetics and pathophysiology of T2D, these results also underscore the value of reanalyzing publicly available data using novel analytical approaches. While this study is of potential interest to the readership of Nat Communication, the study lacks novelty and the replication results of the key signal reported appears to be a singleton variant. I have the following comments:

Comments:

1. It appears from the signal plot in Fig 4a that rs146662075 is a singleton variant with no other SNP in LD supporting the signal. Are there other SNPs in LD that support the association signal that are not shown – otherwise, as this is a low frequency variant, it would need to be sequence confirmed to ensure this is not a spurious signal.
2. What is the MAF of rs146662075 in ExAC in males vs females. Were control datasets used for the association analysis a mixture of males/females or only males
3. Has rs146662075 been reported in any previous association studies with any other phenotypes.
4. While the association signal for rs146662075 is genome wide significant at 3×10^{-8} , the replication signal is not ($P=0.09$) so the signal doesn't meet stringent replication standards despite meta-P being significant at 1.7×10^{-8} . This should be discussed and ideally more expanded replication provided.
5. Permutation analysis at the Chr-X locus would be warranted in further support of the signal for males and females, respectively.
6. Is there supportive data from ENCODE demonstrating functionality beyond pathway clustering (in keeping with the H3K27ac, RNA-seq and luciferase assay data presented)
7. Tissue specific expression of the most significant genes at the Xq23/AGTR2 locus demonstrating enrichment in expression in pancreatic tissue (islet cells) would be helpful information and in particular in the presence of eQTL or differential allelic expression effects. The fetal expression signals presented appear negative in those cell/tissue types; the authors state that they tested the effect of all variants on expression across multiple tissues by interrogating GTEx and RNA-sequencing gene expression data from pancreatic islets - is there any such tissue-specific support from GTEx or other expression databases for the Xq23/AGTR2 locus with respect to pancreas.

8. The manuscript text could be shortened considerably without losing relevant content.

Reviewer #3 (Remarks to the Author):

This paper is another GWAS of T2D line of handful reports, it's novelty relays on the concept of using already available analyzed data, to find, as the authors stated "the value of reanalyzing publicly available data using novel analytical approaches". However, this paper is little bit confusing. While the discovery population is based on published resource the results of the acquired GWAS [it is surprising why meta-analysis was done instead off mega analysis (the individuals' data was available)] have been publically used but not publically published. This confusion continues with the signals that passed genomic significance threshold. It seems like pick and choose strategy, whom to put in either cohort for discovery meta-analysis or the replication meta-analysis in order to pass significant threshold. The question that stems from this analysis is 'why the entire cohorts were not used?' Also, in case of overlap, the discovery cohort should consist of the 70k samples. Only one case (rs2642587) the entire 70k cohort is use but here again something is missing, in this case the entire replication set, although such information exists for the same exact variant at <http://www.type2diabetesgenetics.org/variantInfo/variantInfo/rs2642587> (p=0.831). Further, the variants that the authors focus on i.e. rs146662075 was not replicated, which suggest significance by chance, although multiple comparison adjustment was done. This observation is supported by the following facts: a. the variant is rare (MAF=0.008) which adjacent to the lower cutoff (to be included in the analysis), b. No supported signals around it (like we see in the other candidates), c. located in the middle of gene desert. It seems to me that although follow up studies by the authors were done on this variant the story is not that simple. There is no indication for functional changes in AGTR2 rather association with T2D development, I wonder what are the prevalence of other T2D risk alleles among these 158 risk allele (rs146662075) carriers? Is the variant (rs146662075) in linkage disequilibrium with functional variant within AGTR2? The authors report that the variant was annotated, given only 158 subjects have it, why loci individual sequencing were not perform? Since very little was done to understand the mechanism of this variant and its association to T2D, I would advise the authors to tone down the claim of "significant contribution to the understanding of the molecular basis of T2D".

In my view this paper is somewhat misleading, while I expect to find new variants that were not discovered before due to this novel approach (reanalyzed existing data), in fact we find the same variants published before but with better significance, this only because the supported cohort was picked for the combined meta-analysis.

However, I must acknowledge that the advantage of this approach utilized clean, filtered, QCed, and published variants thus critiques on this critical stages are excessive. However, it is no surprise that the authors replicate prior endeavors as they are based on the same metadata. Finally, I believe enrichment was done on the entire set including the 57 top hits. Since all 57 top hits were associated with T2D previously, it's obvious that we will see the enrichment of pathways involved in insulin response and the results will be skewed. I would like to see the same analysis done without these 57 top hits.

Response to the referees (August, 2017, Nature Communications, NCOMMS-17-09069-T)

Reviewers' comments:

Reviewer #1 (Remarks to the Author):

Bonas-Guarch et al have performed a re-analysis of published GWAS data (available through EGA and dbGAP databases) on T2D of European ancestry including 70,127 individuals (13,857 with T2D) using novel strategies and imputation based upon 1000G and UK10K panels. A key difference to previous meta-analysis attempts using aggregated summary statistics generated independently by different centers, while these authors performed a unified re-analysis of multiple publicly available datasets applying standardized QC methods, genotype imputation and association methods across the whole genome including the X chromosome. After harmonization and QC 70,127 subjects remained (12,931 with T2D) for analyses. The meta-analysis yielded 15,115,281(6,845,408 common, 3,100,848 low-frequency and 5,169,025 rare) variants across cases and controls with a heterogeneity score <0.75). Using both 1000G and UK10K reference panels for imputation clearly improved number of good quality variants including indels (1,357,753). The meta-analysis strategies involved both individual-level genotypes and summary statistics and queries through the www.type2diabetesgenetics.org portal. A gene-set enrichment analysis showed enrichment of genes expressed in pancreas as well as enrichment for active regulatory enhancers. These analyses identified 57 genome-wide significant loci, 7 of which had not been previously reported. They also created a set of 99% credible variants; out of 8,348 credible variants 927 were indels and 105 genome-wide significant; one such example was previously reported intronic variants in the IGF1BP2 gene . for prioritization of causal variants in a locus the VEP and CADD tools were used and interrogated with GTEx. Analysis of the X chromosome not only replicated the DUSP9 gene locus but also suggested that the original association may be driven by a 3-basepair deletion. In addition, a novel signal in males was identified at Xq23 rs 146662075. Imputation for this variant was only accurate in males. Thereafter they attempted to replicate the finding by imputation in SIGMA and INTERACT cohorts as well as de-novo-genotyping in Danish individuals. They also showed that this locus had an allele-specific effect on transactivation on binding using traditional assays, Luciferase and EMSA. Interestingly, only by selecting individuals from the extremes of the distribution of 2 hr glucose during an OGTT they were able to replicate this finding. Other novel loci included a region on 9q34.2 near the ABO gene and a locus on 10q22.1 including the NEUROG3 gene. The authors emphasized that only re-analysis of publicly available data using novel analytical approaches made it possible to generate these novel genetic discoveries.

Comments

This is an interesting study applying novel analytical tools to identify new genetic variants associated with T2D using published data. The results seem convincing and for some replicated applying interesting approaches. The functional studies on rs 146662075 using traditional The paper is well written, supplement structured and messages clear.

The authors should be acknowledged for Figure 1, it is very nice and informative! I would though pay more attention to the following issues.

We appreciate the reviewers for the positive comments

1. The strategy to use “super-controls” based upon 2 hr glucose is really interesting but why do you restrict it to your X-chromosome SNP. Many of the cohorts studied have OGTT data and you could test whether you obtain stronger associations for your new SNPs but also some “old” established like variants in TCF7L2, MTNR1B, SLC30A8 etc. This could serve as proof of principle and replication for the strategy, otherwise we are left with the view that it is only important for your X chrom SNP.

We thank the reviewer for raising this point and agree that it is important to clarify that the use of “super controls” can be an advantage for all the variants and not only for the X chromosome variant. In order to evaluate this, we re-analyzed all the established associated variants, those reported in Supplementary Table 4, after excluding the controls younger than 55 and when possible, also exclusion of pre-diabetics. We added additional columns to the Supplementary Table 4 to show the results obtained with “super controls” analysis.

Overall, for common variants in established GWAS loci, we observed no differences in the absolute log-odds ratio for common variants (diff abs log-effect <0.01). However, for the established low-frequency variant at the *CCND2* locus, applying this filtering of controls results in nearly one order of magnitude improvement of the p-value (p-value (standard controls) = 2.28×10^{-10} vs p-value (supercontrols) = 3.07×10^{-11}) and the effect size (OR (standard controls) = 0.58 vs OR (supercontrols) = 0.56).

This analysis suggests that purifying the controls can improve the quality of the results, especially in low-frequency or rare, and large effect size variants. This could be explained by the fact that in non-pure controls (i. e. controls that are younger than the average age at onset), carriers of variants with a large effect size may have a larger prevalence of T2D, than carriers of variants of smaller effect size. In fact, for a variant with an OR~2.8 like the rs146662075 X chromosome we expect 30 % of carriers to have or develop T2D in the general population, which is exactly what we observe in the Danish cohort where individuals were followed for 11 years.

Related to this point, the filtering of potential pre-diabetic individuals among controls also improved the new replication analysis that further supports the association of the X chromosome rs146662075 variant. For this replication, we used different datasets, including the recently released UK Biobank and Partners Biobank. In these datasets, the exclusion of controls younger than 55 and without family history of T2D, in the absence of glycemic information, also improved the association of the rs146662075 variant, obtaining genome-wide significant results after combining the discovery and replication results (OR=1.95 [1.56-2.45], $p=7.8 \times 10^{-9}$).

We have updated this information in the abstract, results, and supplementary methods.

2. *The most recent paper asking the same question applying novel analytical methods is ref 14. It would be valuable to discuss similarities and dissimilarities between the current ms and ref 14.*

We agree with the reviewer that it is important to discuss the differences between the current manuscript and Fuchsberger et al paper.

There are several differences between our manuscript and Fuchsberger et al paper. First, the sample size of our study (12,931 cases and 57,196 controls) is substantially larger than that of Fuchsberger et al paper (14,297 cases and 32,774 controls). Second, we used two independent reference panels for genotype imputation, 1000G phase 1 and UK10K, which allowed us to obtain, for each variant, the imputation from the reference panel that provided the highest accuracy. In contrast, Fuchsberger et al. used a single reference panel which was sequenced at a lower coverage than UK10K. Third, the analyses for the 70KforT2D project were performed by in the same standardized way, using the same quality control, phasing, imputation and association methods. This avoids large sources of heterogeneity, in contrast with Fuchsberger et al. study where the different steps of the analysis were performed using different methods and in different institutions before they were shared to meta-analyze them. Fourth, *indels* were not imputed in the Fuchsberger et al paper, while in the 70KforT2D project we imputed with good quality 1,357,753 indels. Finally, the X chromosome was not analyzed in the Fuchsberger et al paper, where we actually found the larger effect signal of our study.

We have now modified the Discussion section accordingly (Page 18).

3. *It would also be valuable to test the novel program LINSIGHT (Nat Gen 49,618,2017) which is supposed to outweigh all other programs for identifying human non-coding variants, better than CADD used in this ms.*

We agree with the reviewer that it is important to evaluate the functional impact of the associated variants with LINSIGHT, which is perhaps, the most state of the art method to predict the impact of non-coding variants.

We thank the reviewer for suggesting this recent method, as it can complement the CADD results. We have now applied LINSIGHT and obtained a score for all the variants in all the credible sets identified. We included these results in the new Supplementary Table 9. We also updated Figure 4 in order include the LINSIGHT track, where it shows that the variant rs146662074 falls inside a peak of high functional region in chromosome X. In fact, the score obtained in the 10bp region surrounding the variant (0.20) is as high as the average transcription factor binding sites, in agreement with our experimental results that point this variant as part of a TFBS.

We updated this information in Supplementary Table 9 and in the main text (Results, page 10 and page 16), and Online Supplementary Methods, page 30.

4. *I did not find any comment on ethics and IRB approvals. I know that this has been a prerequisite for depositing data in data-bases as dbGap, but many of the data bases used, DIAGRAM, InterAct (not INTERACT) are not primary databases but aggregates of a number of data bases with different IRB approvals. A statement on how this has been secured would be needed.*

We agree with the reviewer that this point should be clarified. For the DIAGRAM study we just used the final meta-analysis summary statistics, for which we do not need IRB approval. We noticed that the IRB approvals were missing for InterAct. We added the following statement with respect to the IRB approvals in the Supplementary Material, Section 3, Page 16: *“Participants in epic interact provided informed consent. This study was approved by each centre ethics committee and the International Agency for Research on Cancer, the coordinating body for EPIC Europe.”*

Reviewer #2 (Remarks to the Author):

Reviewer Critique

In their study, entitled “A comprehensive reanalysis of publicly available GWAS datasets reveals an X chromosome rare regulatory variant associated with high risk for type 2 diabetes”, Dr. Sílvia Bonàs-Guarch and colleagues performed reanalysis of publicly available GWAS data to gain insights into the genetics of diabetes (T2D). The authors accessed GWAS data from 70,127 subjects, and used an innovative imputation and association strategy that allowed them to replicate and fine map 50 known T2D loci, and identify seven novel associated regions: five driven by common variants in or near LYPLAL1, NEUROG3, CAMKK2, ABO and GIP genes; one by a low frequency variant near EHMT2. One of the locus was driven by a rare variant in chromosome Xq23, associated with a 2.7-fold increased risk for T2D in males, and located within an active enhancer associated with the expression of Angiotensin II Receptor type 2 gene (AGTR2), a known modulator of insulin sensitivity. The authors further show that the risk T allele reduces binding of a nuclear protein, resulting in increased enhancer activity in muscle cells. The authors conclude that beyond providing novel insights into the genetics and pathophysiology of T2D, these results also underscore the value of reanalyzing publicly available data using novel analytical approaches. While this study is of potential interest to the readership of Nat Communication, the study lacks novelty and the replication results of the key signal reported appears to be a singleton variant. I have the following comments:

Comments:

1. *It appears from the signal plot in Fig 4a that rs146662075 is a singleton variant with no other SNP in LD supporting the signal. Are there other SNPs in LD that support the association signal that are not shown – otherwise, as this is a low frequency variant, it would need to be sequence confirmed to ensure this is not a spurious signal.*

We understand the concerns of the referee and have now further clarified this point in the manuscript (Results, Page 13). We actually do find variants in LD that support the

association at rs146662075. Figure 4 shows the regional plot of the association at the X chromosome. As shown in this plot, and in the Supplementary Table 5, there are two variants in LD with the top variant. rs139246371 (chrX:115329804, OR= 1.65, $p=3.5 \times 10^{-5}$) is in $r^2=0.37$ with the top. rs6603744 (chrX:115823966, OR=1.28, $p=1.7 \times 10^{-4}$) is in $r^2=0.1$.

2. *What is the MAF of rs146662075 in ExAC in males vs females. Were control datasets used for the association analysis a mixture of males/females or only males*

We checked the frequency in males and females in the Genome Aggregation Database (gnomAD, <http://gnomad.broadinstitute.org/>), which is, to our knowledge the largest aggregation of sequenced genomes. According to the gnomAD database, the frequency in males in Europeans is 0.0091, whereas the frequency in females is 0.012. For the association analyses we used only males (cases and controls).

3. *Has rs146662075 been reported in any previous association studies with any other phenotypes.*

Following the referee's concern, we have searched for additional information through the literature and in databases and found no other association for this variant with any other trait. This is probably because, as we mention in the discussion, the X chromosome is excluded from most GWAS studies. In addition, the fact that this variant is only well imputed by the UK10K reference panel, makes it more unlikely to find it in other studies. But we have also done new association tests to shed some light on this point. We explored the association of this variant against up to 22 different phenotypes included in the GERA cohort and found no significant signals. We also tested if there were differences in the presence of comorbid conditions between carriers and non-carriers of rs146662076 T allele among T2D individuals. Here we found that carriers of the T risk allele for T2D had increase comorbidity of hypertension (OR=5.87, $p=0.026$). Despite these suggestive results, we think that this is out of the scope of this study and would require confirmation with additional replication analyses.

4. *While the association signal for rs146662075 is genome wide significant at 3×10^{-8} , the replication signal is not ($P=0.09$) so the signal doesn't meet stringent replication standards despite meta-P being significant at 1.7×10^{-8} . This should be discussed and ideally more expanded replication provided.*

We agree with the reviewer that it is crucial to have a more robust replication for this particular variant. To obtain a more robust replication, we analyzed the GWAS data from Partners Healthcare Biobank, and the latest release of UK Biobank, which consists on phenotypic and genotypic data from up to ~500,000 individuals.

For each of the cohorts, we phased the X chromosome, imputed with UK10K and tested for association between the rs146662075 (only in males) and T2D. The Partners Biobank dataset (2173 controls, 871 cases) did not show a significant association, possibly due

insufficient power, but the direction was consistent with the discovery results (OR=1.85, p=0.3).

In the UK Biobank (8247 cases and 81472 controls), the rs146662075 variant was significantly associated with T2D (OR=1.51, p=0.01). We note that the effect size is not as large as the effect size observed in the discovery cohorts. This is likely to be due to the possibly unclassified T2D cases and/or presence of pre-diabetics, which could not be excluded due to the lack of any glycemic measure, such as fasting glucose, glycated haemoglobin or oral glucose tolerant test, usually used to identify T2D cases or pre-diabetic individuals among the controls.

After adding these two datasets, the combined association in the replication studies was significant and with a direction of effect that was consistent with the discovery results (OR=1.57, p=0.0014).

Finally, the combined analysis of the discovery and replication cohorts resulted in genome-wide significant results (OR=1.95 [1.56-2.45], p=7.8x10⁻⁹), which actually improves our original signal, and provides a robust association for this variant.

We have updated this information in the abstract, results (pages 14-16), and Supplementary methods (page 19), Figure 1, Figure 3, Supplementary Figure 9.

5. *Permutation analysis at the Chr-X locus would be warranted in further support of the signal for males and females, respectively.*

We agree with the referee that a permutation analysis could provide support to the ChrX signal in the absence of convincing replication results. Given the robust replication that we have now obtained with the additional datasets (see above), we believe that this computationally costly analysis is therefore not needed at this stage.

6. *Is there supportive data from ENCODE demonstrating functionality beyond pathway clustering (in keeping with the H3K27ac, RNA-seq and luciferase assay data presented)*

Besides the enrichment of T2D variants within pathways related with the pathophysiology of T2D and within active enhancers in pancreatic islets, for the particular X chromosome variant, there is additional data from ENCODE that suggest that the rs146662075 variant is located within an active enhancer in muscle cells. This can be observed in Figure 4, which shows that there are H3K27ac marks specifically in fetal muscle cells. The presence of this enhancer in muscle cells was confirmed by *in vitro* luciferase and EMSA assays. In addition, these assays show that the T-allele reduces the binding activity of a nuclear protein, which results in increased enhancer activity.

7. *Tissue specific expression of the most significant genes at the Xq23/AGTR2 locus demonstrating enrichment in expression in pancreatic tissue (islet cells) would be helpful*

information and in particular in the presence of eQTL or differential allelic expression effects. The fetal expression signals presented appear negative in those cell/tissue types; the authors state that they tested the effect of all variants on expression across multiple tissues by interrogating GTEx and RNA-sequencing gene expression data from pancreatic islets - is there any such tissue-specific support from GTEx or other expression databases for the Xq23/AGTR2 locus with respect to pancreas.

In order to clarify this important point raised by the reviewer, we analyzed additional data from ENCODE, as well as data from pancreatic islets. We observed that the enhancer surrounding the rs146662075 variant is not active in pancreatic islets but only in fetal muscle, according to ENCODE data. However, there is no sufficient data to disentangle unequivocally what is the target gene for this enhancer.

As the reviewer states, we interrogated the association of all variants tested in GTEx across multiple tissues. However, the rs146662075 variant could not be analyzed as it was not imputed in this dataset. Future releases of GTEx that include whole genome sequencing data may have this variant genotyped, although we fear that this dataset could be still insufficiently powered to derive direct conclusions. This is also the case for RNA-sequencing data from pancreatic islets, as the largest dataset was based on genotyping with a GWAS array and imputation with 1000G, which cannot capture accurately the rs146662075 variant, as compared with UK10K reference panel.

To identify the target gene regulated by this enhancer, we analyzed the correlation between gene expression and enhancer activity (measured by H3K27ac) across 24 tissues available through Roadmap Epigenomics. As shown in Figure 4 and supplementary Figure 11, we observed that the presence of active enhancer was significantly associated with the presence of expression of *AGTR2*, but not with the expression of the other genes in the topologically associated domain (TAD) ($p[\text{Fisher's test}]=0.0014$). We cannot exclude that other genes in the same TAD, i. e. *LOC642776*, *SLC6A14*, or *KLHL13* could also be regulated by this enhancer in a particular tissue or cell type not included within ENCODE.

We have added these statements to the Discussion Section (page 16).

8. The manuscript text could be shortened considerably without losing relevant content.

Following the reviewer's suggestion, we have now shortened the overall length of the manuscript substantially, despite adding more data and results.

Reviewer #3 (Remarks to the Author):

This paper is another GWAS of T2D line of handful reports, it's novelty relays on the concept of using already available analyzed data, to find, as the authors stated "the value of reanalyzing publicly available data using novel analytical approaches". However, this paper is little bit confusing. While the discovery population is based on published

resource the results of the acquired GWAS [it is surprising why meta-analysis was done instead of mega analysis (the individuals' data was available)] have been publically used but not publically published.

We apologize for not clarifying this further in the manuscript. There are several reasons why we believe that a meta-analysis is more appropriate for this study. One of the reasons is that each of the cohorts has been ascertained using different selection criteria and genotyped with different arrays, which would make mega-analysis very challenging, especially due to the differences in imputation qualities for each of the variant across datasets. Therefore, we believe that a meta-analysis can account better for the between study differences. In addition, merging the data from all the datasets in a single dataset is also computationally very challenging and increases the chances of generating undesired artifacts. For all these reasons we chose to use a meta-analysis strategy, which maximizes the power and accuracy.

We would also like to clarify that the results of this study are currently publicly available through the Type 2 Diabetes Knowledge Portal (<http://www.type2diabetesgenetics.org/>)

This confusion continues with the signals that passed genomic significance threshold. It seems like pick and choose strategy, whom to put in either cohort for discovery meta-analysis or the replication meta-analysis in order to pass significant threshold. The question that stems from this analysis is 'why the entire cohorts were not used?' Also, in case of overlap, the discovery cohort should consist of the 70k samples.

We again apologize for not making this point clear enough in our manuscript. We have now further clarified our general strategy in the manuscript. Briefly, the DIAGRAM dataset contained only a fraction (1.9M of variants) of all the variants analyzed in the 70KforT2D dataset (~15M of variants). For all of the variants that were present in the DIAGRAM dataset we meta-analyzed the results of the DIAGRAM dataset with the 70K dataset. However, three of the cohorts in the 70K dataset overlap with cohorts in the DIAGRAM dataset. Since we could not exclude cohorts from the DIAGRAM dataset, as only the meta-analysis summary statistics were available, we had to exclude cohorts from the 70K dataset in order to avoid overlap while taking advantage of the increased power achieved by adding the DIAGRAM data.

We have reviewed the text in the Results section (page 7), to try to make this strategy more clear.

Only one case (rs2642587) the entire 70k cohort is use but here again something is missing, in this case the entire replication set, although such information exists for the same exact variant at <http://www.type2diabetesgenetics.org/variantInfo/variantInfo/rs2642587> (p=0.831).

The rs2642587 variant is represented by several datasets in the type 2 diabetes knowledge portal (<http://www.type2diabetesgenetics.org/>), including the association data presented in this manuscript (70KforT2D) that is currently also shared through the T2D

portal. This variant is nominally significant, and in consistent direction in the GoT2D WGS + replication dataset (OR=1.07, p=0.02, 14,297 cases and 32,774 controls). As the reviewer points, that variant is not significant in the SIGMA dataset (p=0.8, N=8,891 samples), which is expected, given the fact that the SIGMA dataset is almost one order of magnitude smaller compared to the 70KforT2D dataset and therefore underpowered to identify such association.

Further, the variants that the authors focus on i.e. rs146662075 was not replicated, which suggest significance by chance, although multiple comparison adjustment was done. This observation is supported by the following facts:

- a. the variant is rare (MAF=0.008) which adjacent to the lower cutoff (to be included in the analysis),*
- b. No supported signals around it (like we see in the other candidates),*
- c. located in the middle of gene desert.*

We agree with the referee that the replication of this variant required additional support. To obtain a more robust replication, we analyzed the GWAS data from Partners Healthcare Biobank, and the latest release of UK Biobank, which consists on phenotypic and genotypic data from up to ~500,000 individuals.

For each of the cohorts, we phased the X chromosome, imputed with UK10K and tested for association between the rs146662075 (only in males) and T2D. The Partners Biobank dataset (2173 controls, 871 cases) did not show a significant association, possibly due insufficient power, but the direction was consistent with the discovery results (OR=1.85, p=0.3).

In the UK Biobank (8247 cases and 81472 controls), the rs146662075 variant was significantly associated with T2D (OR=1.51, p=0.01). We note that the effect size is not as large as the effect size observed in the discovery cohorts. This is likely to be due to the possibly unclassified T2D cases and/or presence of pre-diabetics, which could not be excluded due to the lack of any glycemic measure, such as fasting glucose, glycated haemoglobin or oral glucose tolerant test, analytic measures usually used to identify T2D cases or pre-diabetic individuals among the controls.

After adding these two datasets, the replication sample was significant and with a direction of effect that was consistent with the discovery results (OR=1.57, p=0.0014).

Finally, the combined analysis of the discovery and replication cohorts resulted in genome-wide significant results (OR=1.95 [1.56-2.45], p=7.8x10⁻⁹), which actually improves our original signal, and provides a robust association for this variant.

We have updated this information in the abstract, results (pages 14-16), and Supplementary methods (page 19), Figure 1, Figure 3, and Supplementary Figure 9.

Furthermore, in response to the three points raised by the referee as indicative of false positive signal, we would like to state that (a) our lower MAF cutoff for including a variant in the study is 0.001. This variant has a frequency of 0.008, which we do not think is adjacent to the lower cutoff, given the distribution of the MAFs across the cohorts. (b) Although we would not expect to find additional signals in LD for rare variants, as often as for common variants, we actually do find two other variants in LD with rs146662075 that support the association signal. Figure 4 shows the regional plot of the association at the X chromosome. As shown in this plot, and in Supplementary Table 5, there are two variants in LD with the top variant: rs139246371 (chrX:115329804, OR= 1.65, $p=3.5 \times 10^{-5}$) and rs6603744 (chrX:115823966, OR=1.28, $p=1.7 \times 10^{-4}$), which are in r -squared of 0.37 and 0.1 with the top variant, respectively. Finally, (c) it is quite common to find variants associated to complex diseases in intergenic regions, as it has been reported for many common diseases, including for T2D (i. e. Pasquali L et al. Nat Genet. 2014).

It seems to me that although follow up studies by the authors were done on this variant the story is not that simple. There is no indication for functional changes in AGTR2 rather association with T2D development, I wonder what are the prevalence of other T2D risk alleles among these 158 risk allele (rs146662075) carriers?

This is a very good question that we did not completely explore within our original study. In order to disprove that the association of rs146662075 variant is due to the presence of other risk alleles in the carriers, we performed conditional analysis with the four variants that showed larger effect size and genome-wide significance in our dataset. If the association was in fact caused by the presence of additional T2D risk alleles we would expect the association at rs146662075 to disappear when conditioning on these variants. We therefore performed the association analysis again, but conditioning by the following variants:

KCNQ1-INT15 (rs2237897, Unoki et al. 2008)
CCND2 (rs76895963, Steinthorsdottir et al. 2014)
PPARG (rs1801282, Morris et al. 2012)
TCF7L2 (rs7903146, Voight et al. 2010)

From this analysis, we observed that the signal remains significant with similar values: OR=2.84 and $p=9.18 \times 10^{-8}$, versus original discovery results (OR=2.94, $p=3.52 \times 10^{-8}$), finally concluding that the signal does not derive from these other loci.

Is the variant (rs146662075) in linkage disequilibrium with functional variant within AGTR2? The authors report that the variant was annotated, given only 158 subjects have it, why loci individual sequencing were not perform? Since very little was done to understand the mechanism of this variant and its association to T2D, I would advise the authors to tone down the claim of “significant contribution to the understanding of the molecular basis of T2D”.

The rs146662075 variant is not, according to current linkage disequilibrium and functional annotation maps, linked to functional coding variants in *AGTR2*. Furthermore, the variant is carried by more than 158 subjects. In fact, in total, 191 cases and 505 controls contain the variant when considering the discovery and replication together. The variant has been imputed in the majority of the cohorts, but, as an alternative to sequencing, we have also directly genotyped it within the Danish cohort, where we observe that carriers have three-fold risk of developing incident diabetes after 11 years of follow-up (Figure 3, Results, page 14).

As the reviewer suggests, we toned down the referred claim in the discussion.

In my view this paper is somewhat misleading, while I expect to find new variants that were not discovered before due to this novel approach (reanalyzed existing data), in fact we find the same variants published before but with better significance, this only because the supported cohort was picked for the combined meta-analysis.

Again, we apologize if this point was not clear enough in the manuscript. The reanalysis of publicly available data, combined with improved and more rigorous methodology allowed us to identify 50 already known loci, and seven other genome-wide significant association signals that, to our knowledge, were not previously reported and are therefore new. We would also like to further clarify that the strategy for combining cohorts for the meta-analysis was designed aiming at maximizing the signal while avoiding data overlap, and adapting to their different levels of usage restrictions. For example, the results from the DIAGRAM trans-ethnic meta-analysis, from which we could access only the summary statistics, were chosen when that variant was available, as this resulted in the largest sample size and statistical power. However, this was only possible for around 1.9M HapMap variants that were analyzed in this study.

However, I must acknowledge that the advantage of this approach utilized clean, filtered, QCed, and published variants thus critiques on this critical stages are excessive. However, it is no surprise that the authors replicate prior endeavors as they are based on the same metadata.

Finally, I believe enrichment was done on the entire set including the 57 top hits. Since all 57 top hits were associated with T2D previously, it's obvious that we will see the enrichment of pathways involved in insulin response and the results will be skewed. I would like to see the same analysis done without these 57 top hits.

We apologize once more for not making this point clear in the manuscript. We would like to clarify that the main goal of this study was to identify novel associated regions, as well as fine-map the existing ones. Of all the 57 genome-wide hits that we identified, 7 of them have not been previously reported and are therefore considered novel. As a confirmation that our results captured processes related with T2D, we performed two types of enrichment analysis. First, we used DEPICT to search for pathways or gene-sets that are enriched in genes that harbor T2D associated variants. This resulted in enrichment for genes expressed in pancreas, as well as genes involved in cellular response to insulin stimulus. This analysis used all the variants (genome-wide) with p-

value below 1×10^{-5} and therefore used more data than the data from these 57 genome-wide significant hits.

As an alternative approach, we used variant set enrichment analysis to identify enrichment of regions with active enhancers in pancreatic islets. This enrichment used the 57 genome-wide significant hits, of which 7 are novel. While the DEPICT results have been reported here for the first time, the enrichment of active enhancers have been previously reported, as we acknowledge in the results section (Result section, page 8). However, we believe that it is still of value to replicate the previous observations and we did not derive any further conclusion from them, other than confirming that the association results indeed captured processes related with T2D. We did not derive any further conclusion from them, as we believe that this type of test is only suggestive. We could not perform the analysis without the 50 already known loci, as we did not have enough power to detect association with only 7 GWAS hits.

Reviewer #1 (Remarks to the Author):

I am pleased with the responses by the authors and changes in manuscript. No further comments.

Reviewer #2 (Remarks to the Author):

The authors have been responsive to the comments raised by the reviewers and have done a good job revising the manuscript. The additional support provided for the X-chr locus association now leaves this as a bonofide signal that is novel. I have no further edits/suggestions on the manuscript.

Reviewer #3 (Remarks to the Author):

In my opinion, the use of the UK and Partners Biobanks was a tiebreaker and indeed greatly improve both the message and the coherence of the report.
No further comments.